# Ligand recognition and G protein coupling of the human itch receptor MRGPRX1

Lulu Guo[1,2,14], Yumu Zhang[3,4,14], Guoxing Fang[5,14], Lu Tie[6,14], Yuming Zhuang[5,14], Chenyang Xue[7,14], Qi Liu[1,14], Minghui Zhang[2], Kongkai Zhu[2], Chongzhao You[3], Peiyu Xu[3], Qingning Yuan[3], Chao Zhang[2], Lei Liu[1], Naikang Rong[2], Shengxuan Peng[1], Yuan Liu[8], Chuanzheng Wang[1], Xin Luo[1], Zongyao Lv[1], Dongwei Kang[9], Xiao Yu[5], Cheng Zhang[8], Yi Jiang[10], Xinzhong Dong[11,12], Jiuyao Zhou[13]✉, Zhongmin Liu[7]✉, Fan Yang[1,5]✉, H. Eric Xu[3]✉ & Jin-Peng Sun[1,2,6]✉

MRGPRX1, a Mas-related GPCR (MRGPR), is a key receptor for itch perception and targeting MRGPRX1 may have potential to treat both chronic itch and pain. Here we report cryo-EM structures of the MRGPRX1-Gi1 and MRGPRX1-Gq trimers in complex with two peptide ligands, BAM8-22 and CNF-Tx2. These structures reveal a shallow orthosteric pocket and its conformational plasticity for sensing multiple different peptidic itch allergens. Distinct from MRGPRX2, MRGPRX1 contains a unique pocket feature at the extracellular ends of TM3 and TM4 to accommodate the peptide C-terminal "RF/RY" motif, which could serve as key mechanisms for peptidic allergen recognition. Below the ligand binding pocket, the $G^{6.48}XP^{6.50}F^{6.51}G^{6.52}X_{(2)}F/W^{6.55}$ motif is essential for the inward tilting of the upper end of TM6 to induce receptor activation. Moreover, structural features inside the ligand pocket and on the cytoplasmic side of MRGPRX1 are identified as key elements for both Gi and Gq signaling. Collectively, our studies provide structural insights into understanding itch sensation, MRGPRX1 activation, and downstream G protein signaling.

Itch and its scratch responses are normal physiological processes of many animals. Only after the bony fish and tetrapods diverged could animals scratch with their arms and legs to remove parasites or other unwanted stimuli on the skin[1,2]. Primary sensory neurons in the dorsal root ganglion (DRG) play important roles in detecting, transmitting, and modulating sensory signals including itch and pain from the periphery (skin) to the spinal cord[3]. To establish an efficient detection-scratching response cycle, tetrapods have developed a family of seven

[1]Advanced Medical Research Institute, Cheeloo College of Medicine, Shandong University, Jinan, China. [2]Key Laboratory of Experimental Teratology of the Ministry of Education, Department of Biochemistry and Molecular Biology, Shandong University School of Medicine, Jinan, China. [3]State Key Laboratory of Drug Research, Shanghai Institute of Materia Medica, Chinese Academy of Sciences, Shanghai, China. [4]School of Life Science and Technology, ShanghaiTech University, Shanghai 201210, China. [5]Key Laboratory Experimental Teratology of the Ministry of Education and Department of Physiology, School of Basic Medical Sciences, Shandong University, Jinan, Shandong, China. [6]Department of Pharmacology, School of Basic Medical Sciences, Peking University and Beijing Key Laboratory of Tumor Systems Biology, Peking University, Beijing, China. [7]Department of Immunology and Microbiology, Southern University of Science and Technology, Shenzhen, Guangdong, China. [8]Qilu Hospital, Cheeloo College of Medicine, Shandong University, Jinan, Shandong, China. [9]Department of Medicinal Chemistry, Key Laboratory of Chemical Biology (Ministry of Education), School of Pharmaceutical Sciences, Shandong University, Jinan, China. [10]Lingang Laboratory, Shanghai, China. [11]The Solomon H. Snyder Department of Neuroscience, Johns Hopkins University School of Medicine, Baltimore, MD, USA. [12]Howard Hughes Medical Institute, Johns Hopkins University School of Medicine, Baltimore, MD, USA. [13]Department of Pharmacology, School of Pharmaceutical Sciences, Guangzhou University of Chinese Medicine, Guangzhou, China. [14]These authors contributed equally: Lulu Guo, Yumu Zhang, Guoxing Fang, Lu Tie, Yuming Zhuang, Chenyang Xue, Qi Liu. ✉e-mail: zhoujiuyao@tom.com; liuzm@sustech.edu.cn; yangfan1357@163.com; eric.xu@simm.ac.cn; sunjinpeng@sdu.edu.cn

transmembrane receptors, many of which are Mas-related GPCRs (MRGPRs) expressed in the DRG[4], to sense itch and report the sensations to the central nervous system, generating scratch or avoidance behaviors[1]. However, persistent and unsolved itch/pruritus is usually an unpleasant disease that may result in depression and various mental illnesses in the case of severe conditions[5]. Therefore, it is desirable to understand the molecular mechanisms governed by MRGPRs that underlie itch signaling in both physiological and pathological conditions, which may facilitate therapeutic development to treat pruritus.

Several of the MRGPRs, including MRGPRX1 and MRGPRX4 in humans and MrgprC11, MrgprA1, and MrgprA3 in rodents, exist in primary sensory neurons to serve as sensors for itchy stimuli, such as pruritogen peptides, exogenous chemicals, or clinical drugs that have allergic side effects[6]. In particular, MRGPRX1, one of the itch receptors in humans, is able to respond to endogenous peptides, such as γ2-melanocyte-stimulating hormone (γ2-MSH)[7], bovine adrenal medulla peptide (BAM) 8-22[8], and the antimalarial drug chloroquine (CQ)[6], which has an intolerant itch side effect.

Downstream of MRGPRX1, both Gi and Gq signaling are activated in response to the endogenous pruritogen BAM8-22 or the exogenous clinical drug CQ[8,9]. The functional homologs of MRGPRX1 in mice, MrgprA3 and MrgprC11, are known to functionally link to TRPA1, which is essential for itch sensation. These two receptors are functionally linked to TRPA1 through different mechanisms. Whereas Mrgprc11 connected to TRPA1 through Gq-PLC signaling, MrgprA3 was found to link to TRPA1 through Gβγ. Using dental afferents of human samples, MRGPRX1 was shown to sensitize TRPA1 and instigate membrane depolarization[10,11]. These results reveal an important signaling network for itch sensation in the peripheral nervous system, initiated by the activation of MRGPRX1.

In addition to evoking itch sensation at the peripheral axons, recent studies suggested that activation of MRGPRX1 located in the central terminals of primary sensory neurons in the spinal cord led to the activation of the Gi1 pathway, inhibition of neurotransmitter release, and attenuated chronic pain[12]. To understand the structural basis of the itch sensation, particularly in the sensation processes of the peripheral nervous system, and to develop important therapeutic tools for the treatment of itch-related diseases and pains in the central nervous system, we determined the structures of MRGPRX1-Gi1 in complex with bovine adrenal medulla 8-22 (BAM8-22) or CNF-Tx2 and the structure of MRGPRX1-Gq in complex with BAM8-22 at the resolutions of 3.0 Å, 2.8 Å, 2.7 Å, respectively.

## Results

### Overall structures of MRGPRX1 complexes

To provide the structural basis for the ligand recognition and activation mechanism of MRGPRX1, we assembled 3 structures of MRGPRX1-Gi1/Gq in complex with two peptide agonists. To increase the expression level of MRGPRX1, we fused a BRIL tag at the N-terminus of full-length wild-type MRGPRX1. Incubation of ligands with membranes from cells co-expressing the receptor and heterotrimeric Gi1 or chimeric Gq protein in the presence of scFv16 enabled the effective assembly of MRGPRX1-Gi1/Gq complexes, which produced highly homogenous complex samples for structural studies (Supplementary Fig. 1).

The two structures of BAM8-22-MRGPRX1 in complexes with Gq or Gi1 trimer were determined by single particle cryo-EM at resolutions of 2.7 Å and 3.0 Å, respectively (Figs. 1a, b and Supplementary Table 1). Moreover, the structure of MRGPRX1-Gi1 bound to venom CNF-Tx2 was determined at a global resolution of 2.8 Å (Fig. 1a, b and Supplementary Figs. 2–4). The EM densities of these structures enabled model building of the overall transmembrane helices (TM1-TM7) and most side chains (Supplementary Fig. 5). Notably, a partial ECL2 segment of MRGPRX1 had weak EM density, and residues M161-W172 were not modeled in the final structures (Supplementary Tables 2, 3, 4).

Inspection of these structures identified important structural information for the binding interface between ligands and the MRGPRX1 receptor, as well as the coupling interface between the receptor and the Gq/Gi1 heterotrimer.

All ligands of MRGPRX1 occupied a shallower ligand pocket than those in most class A GPCRs, with a vertical distance ranging from 12.8 Å to 17.9 Å from the conserved "toggle switch" position of G229[6.48] (Fig. 1c). This observation led us to recall a similar binding mode of peptides or small chemical ligands to another itch receptor member[13], MRGPRX2, indicating that a shallower pocket with significant plasticity serves as one of the common features for agonist recognition by MRGPR itch receptor families.

In general, the overall structure of MRGPRX1 bound with different ligands assumed similar conformations, with root-mean-square-deviation (RMSD) values ranging from 1.06 Å to 1.38 Å among themselves (Supplementary Fig. 6a) and from 1.74 Å to 1.93 Å with our recently solved MRGPRX2 structures (Supplementary Fig. 6b). However, the binding modes of the peptide agonists of MRGPRX1, including BAM8-22 and CNF-Tx2, were distinct from those of MRGPRX2 with a reversed orientation (Fig. 1d). Notably, in the MRGPRX2 complex structures, the binding modes of C14 and PAMP-12 started from TM4 and reached to TM1 and TM2. In contrast, the binding of BAM8-22 and CNF-Tx2 in MRGPRX1 started from TM5-TM6 at the N-terminus and reached TM4 at the C-terminus (Fig. 1d). Importantly, MRGPRX1 and MRGPRX2 showed distinct ligand recognition patterns; for instance, the MRGPRX2 ligand PAMP-12 cannot activate MRGPRX1, and the MRGPRX1 ligand BAM8-22 cannot be recognized by MRGPRX2 (Fig. 1e). These different binding modes of peptide agonists indicated potential distinct structural elements or motifs recognized by different itch receptor members, thus delineating their selective functions.

### Binding of BAM8-22 to MRGPRX1

MRGPRX1 was able to sense endogenous and exogenous peptides that shared a conserved sequence of RF/Y-G or RF/Y-amide near their C-terminal[14,15], exemplifying a distinct sequence feature compared with our recently identified motif $\varphi^{p9}(X_{0-1})$ R/K$^{p10}(X_2)$ $\varphi^{p13}(X_{2-3})$ $\varphi^{p16}(X_3)$ R/K$^{p20}$ recognized by another itch receptor, MRGPRX2[13,14]. BAM8-22 is the C-terminal fragment of the opioid peptide BAM, truncated by the removal of the N-terminal met-enkephalin motif (YGGFM) for efficient opioid binding[15]. In both the BAM8-22-MRGPRX1-Gi1 and BAM8-22-MRGPRX1-Gq complex structures, BAM8-22 assumed a "U"-shaped binding pose, with an RMSD of Cα at 0.94 Å (Fig. 2a and Supplementary Fig. 7a).

BAM8-22 in the MRGPRX1 complex structure provided a typical two-layer binding model, with five hydrophobic residues and one conserved basic residue R[20] buried inside and four polar side chains facing outward toward the solution. Importantly, Y[21] sat in a hydrophobic pocket constituted by Y99[3.29], P100[3.30], E157[4.60], and W158[4.61] (Fig. 2a, b; Supplementary Fig. 7b and Supplementary Tables 5 and 6), and R[20] formed a salt bridge or hydrogen bond network with E157[4.60], D177[5.36], and Y99[3.29] (Fig. 2a, b; Supplementary Fig. 7c and Supplementary Tables 5 and 6). These observations were consistent with the previously identified "RF/RY" motif, which is specific for peptide agonist recognition by MRGPRX1. N-terminal to these two residues, the two hydrophobic residues M[15] and Y[17] sat in a hydrophobic pocket encompassed by F236[6.55], H254[7.35], F250[7.31], R246[ECL3], and L249[7.30] (Fig. 2b; Supplementary Fig. 7d and Supplementary Table 6), and the successive residues W[13] and W[14] lay along a hydrophobic patch created by L240[6.59], W241[6.60], I242[6.61] and H243[6.62] in the BAM8-22-MRGPRX1-Gq complex structure (Fig. 2b; Supplementary Fig. 7e and Supplementary Table 6).

We next used unbiased alanine scanning mutagenesis for each residue in BAM8-22 and the ligand-binding pocket of MRGPRX1 to identify the hotspot interactions between MPRGPRX1 and BAM8-22. Consistent with the previously proposed "RY/RF" motif, mutations of

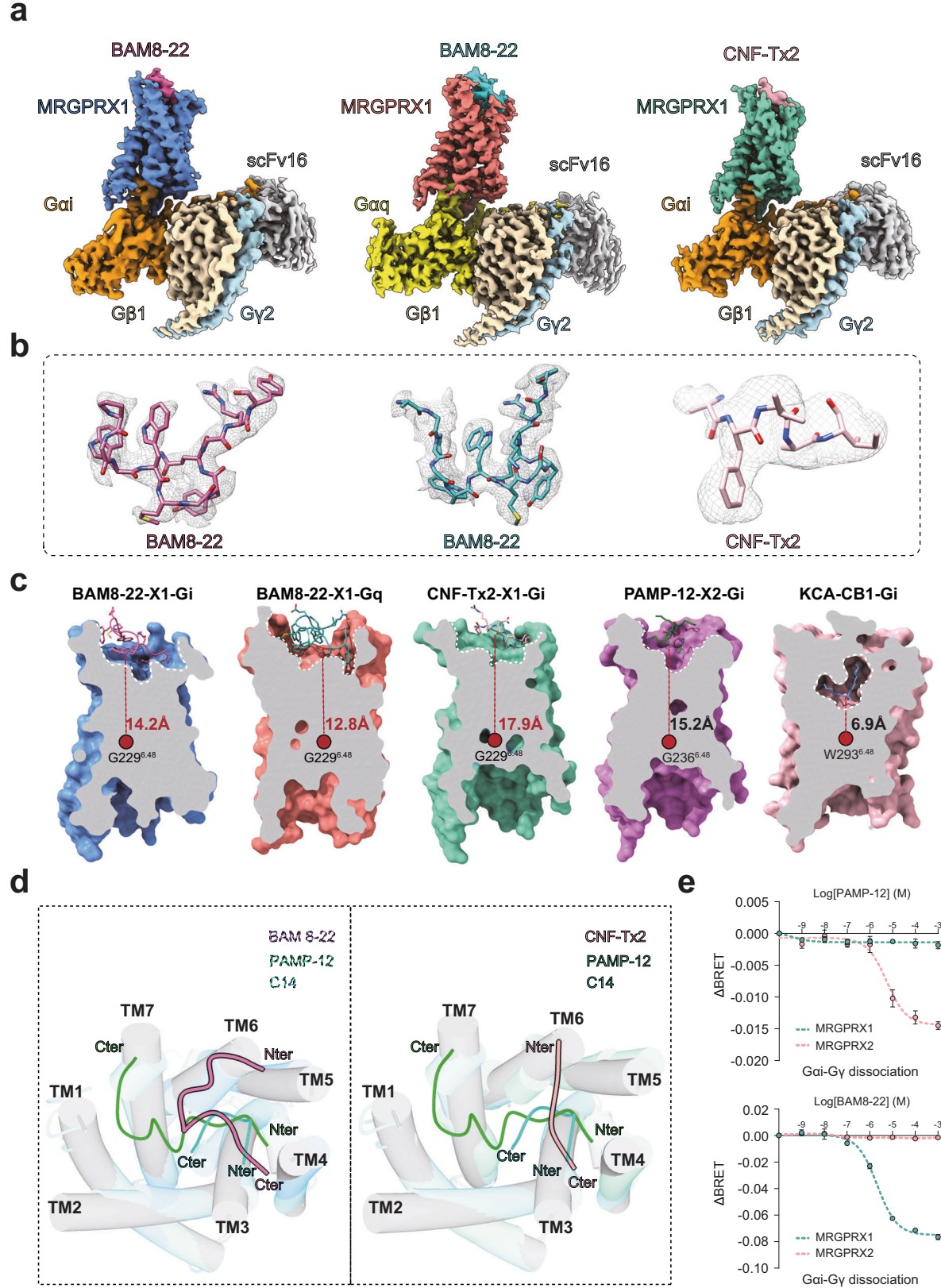

either R[20] or Y[21] significantly decreased the Gq or Gi1-coupling activity of MRGPRX1 (Fig. 2c and Supplementary Fig. 8a, b). Mutations of ten pocket residues surrounding the EM density corresponding to BAM8-22 also markedly reduced the activity of MRGPRX1 in response to BAM-8-22 binding (Fig. 2d and Supplementary Fig. 8c). We next used a BRET-based assay that monitors Gi1-Gβγ dissociation to examine MRGPRX1

signaling in response to the individual BAM8-22 mutants compared with that of the wild-type peptide for the MRGPRX1 pocket mutations associated with a significant loss in binding ability (Fig. 2e). If a mutation in MRGPRX1 severely inhibited the response to wild-type BAM8-22 but had much less serious effects on the response to a BAM8-22 peptide with a particular alanine substitution, then the MRGPRX1 mutation

**Fig. 1 | Cryo-EM structure of MRGPRX1-Gi1/Gq complexes. a** Cryo-EM density map of the BAM8-22-MRGPRX1-Gi1/Gq complexes and CNF-Tx2-MRGPRX1-Gi1 complex. **b** EM density maps of BAM8-22 in MRGPRX1-Gi1/Gq complexes and CNF-Tx2 in MRGPRX1-Gi1 complex. BAM8-22 in MRGPRX1-Gi1/Gq complexes, hot pink/cyan; CNF-Tx2, light pink; MRGPRX1, cornflower blue, salmon, and medium aquamarine from left to right; Gαi, dark orange; Gαq, yellow; Gβ, wheat; Gγ, light blue; scFv16 (single-chain variable fragment), gray. **c** Cut-away view of the ligand-binding pocket in the BAM8-22-MRGPRX1-Gi1/Gq, CNF-Tx2-MRGPRX1-Gi1, PAMP-12-MRGPRX2-Gi1 and KCA-CB1-Gi1 complexes. **d** The binding modes of the peptide agonists of MRGPRX1 compared with MRGPRX2. **e** The selectivity of different ligands for MRGPRX1 and MRGPRX2 was evaluated using Gαi-Gγ dissociation assay. Data are mean ± SEM of three independent experiments (n = 3). ΔBRET indicates the difference between the BRET signal recorded from cells treated with ligand and that from cells treated with vehicle.

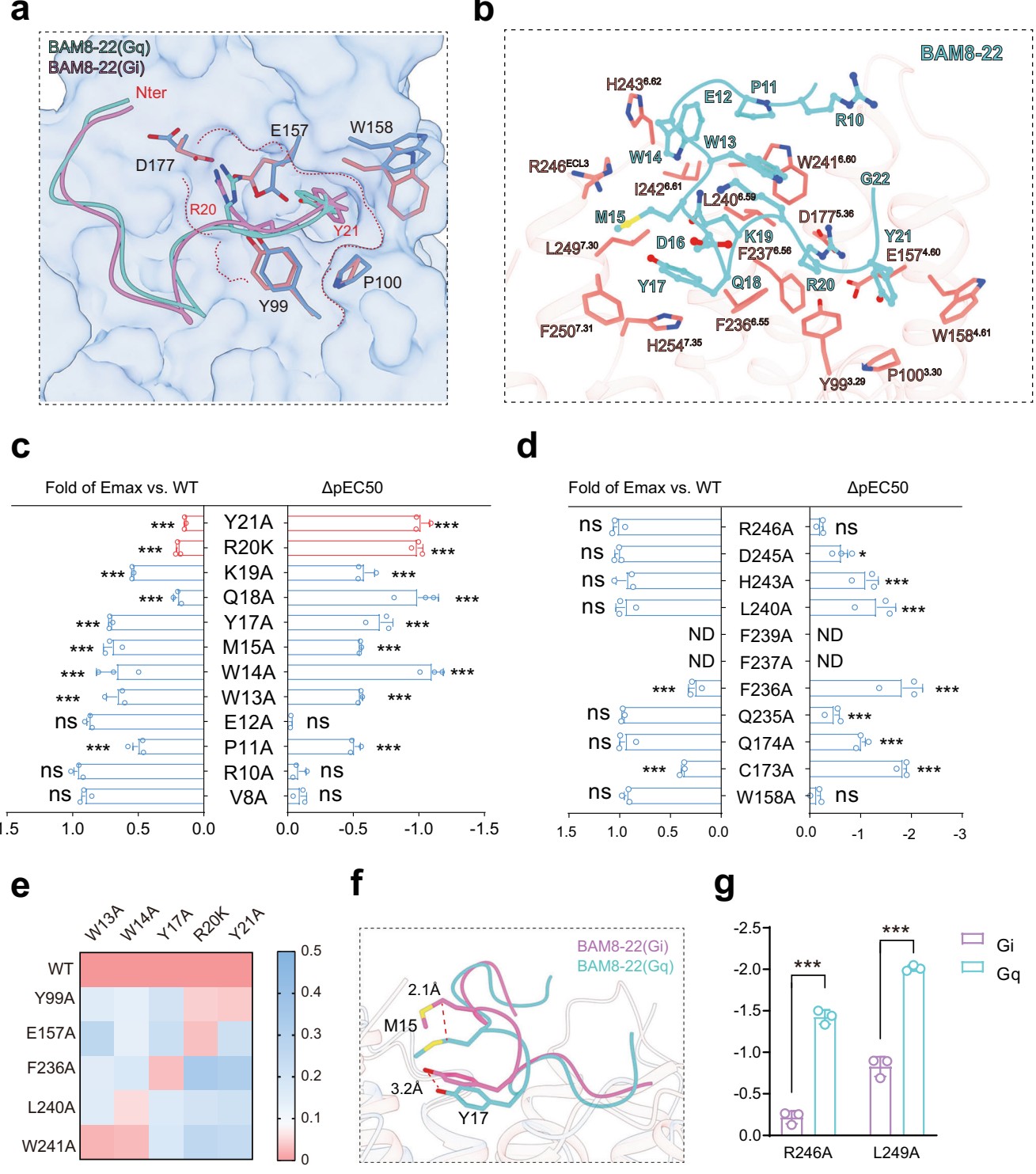

**Fig. 2 | The BAM8-22 binding pocket in MRGPRX1-Gi1/Gq complexes. a** The "U"-shaped binding pose of BAM8-22 in MRGPRX1-Gi1/Gq complexes. **b** Three-dimensional (3D) representation of the detailed interactions between BAM8-22 and MRGPRX1 in MRGPRX1-Gq complex. **c** Effects of different BAM8-22 mutations on BAM8-22 induced Gαi-Gγ dissociation. A bar graph for EC50 was presented. Due to the poor solubility of peptide BAM8-22-R20A, we used BAM8-22-R20K for the activity pairing assays of BAM8-12 mutants. Statistical differences between BAM8-22 WT and mutations were determined by two-sided one-way ANOVA with the Tukey test. *$P < 0.05$; **$P < 0.01$; ***$P < 0.001$; ns, no significant difference. ($P = $ <0.001, <0.001, <0.001, <0.001, <0.001, <0.001, <0.001, <0.001, 0.9999, <0.001, 0.7619, 0.6286 from top to bottom of ΔpEC50; $P = $ <0.001, <0.001, <0.001, <0.001, <0.001, <0.001, <0.001, <0.001, 0.1359, <0.001, 0.9905, 0.3709 from top to bottom of Emax). Data from three independent experiments are presented as the mean ± SEM ($n = 3$). **d** Effects of different mutations within the ligand-binding pocket of MRGPRX1 on BAM8-22 induced Gαi-Gγ dissociation. Statistical differences between MRGPRX1 WT and mutations were determined by two-sided one-way ANOVA with the Tukey test. *$P < 0.05$; **$P < 0.01$; ***$P < 0.001$; ns, no significant difference. ($P = 0.7609$, 0.0117, <0.001, <0.001, ND, ND, <0.001, 0.0003, <0.001, <0.001, 0.9616 from top to bottom of ΔpEC50; $P = 0.9916$, 0.9642, 0.1755, 0.2536, ND, ND, <0.001, 0.4812, 0.2536, <0.001, 0.1643 from top to bottom of Emax). Data from three independent experiments are presented as the mean ± SEM ($n = 3$). **e** Heatmap of the pairing of BAM8-22 mutants with MRGPRX1 WT and MRGPRX1 alanine scanning mutants. The receptor mutants that did not show significantly decreased EC50 values compared to those of the WT receptor when binding to a specific BAM8-22 mutant are highlighted by red color. **f** Structural representation of M15 and Y17 in MRGPRX1-Gαi/Gq complexes, respectively. The distance was depicted as the dashed red line. **g** Effects of F236[6.55], R246[ECL3,] and L249[7.30] on BAM8-22 induced Gαi/Gq activity by Gαi-Gγ dissociation assay. The curve data from three independent measurements are measured as mean ± SEM ($n = 3$). All data were determined by two-sided one-way ANOVA with the Tukey test. ***$P < 0.001$, ns, no significant difference.

and BAM8-22 alanine substitution could be paired together[16]. The mutagenesis scanning data indicated that the effect of E157A of MRGPRX1 paired well with the effect of R20K of BAM8-22, which was consistent with the strong salt bridge observed in the BAM8-22-MRGPRX1-Gi1 complex structure (Fig. 2e). Whereas Y17 of BAM8-22 functionally paired with F236[6.55] of MRGPRX1, W13, and W14 of BAM8-22 paired well with W241[6.60] of MRGPRX1. These biochemical data supported that the key interactions, including the salt bridges and hydrophobic packings mediated by the R20-E157[4.60], Y17-F236[6.55,] and W13/W14-W241[6.60] pairs, played central roles in the recognition of BAM8-22 by MRGPRX1 (Fig. 2e).

In the BAM8-22-MRGPRX1-Gi1 complex structure, M15 and Y17 assumed a shallower position, sitting approximately 2 Å higher than the corresponding residues in the BAM8-22-MRGPRX1-Gq complex structure (Fig. 2f). The conformational changes of M15 and Y17 decreased their interactions with R246[ECL3] and L249[7.30] (Fig. 2b and Supplementary Fig. 7f). The different interaction modes of BAM8-22 with MRGPRX1 in different G-protein subtype coupling states may contribute to its selectivity for Gi1 or Gq engagement. Consistently, mutational studies indicated that R246[ECL3] and L249[7.30] are more important for BAM8-22-induced Gq activity than for Gi activity (Fig. 2g and Supplementary Fig. 8d). Notably, among residues whose mutations showed effects on BAM8-22 induced MRGPRX1 activity, several of them, such as E12, K19 in BAM8-22-MRGPRX1-Gi complex don't have unambiguous side chain assignment. Therefore, direct interactions between these residues of BAM8-22 and MRGPRX1 were not observed. Moreover, there may be other residues of BAM8-22 that contributed more significantly to the entropy changes, but not enthalpy changes, for the binding of BAM8-22 to MRGPRX1.

## Binding of CNF-Tx2 to MRGPRX1

The CNF-Tx2, an RF-amide peptide derived from the C. textile venom gland, is a known itch allergen and MRGPRX1 agonist[17]. Putative EM densities fitting part of the sequence of CNF-Tx2 could be located in the CNF-Tx2-MRGPRX1-Gi1 complex structure but could not be unambiguously assigned, which may be due to the multiple potential binding modes involving the repeated presence of the core sequence R14F15/R17I18 of CNF-Tx2[6] (Fig. 3a). We then fit partial sequences of CNF-Tx2 with two different models into the EM density, one with residues of F15-R17I18 (model 1) and the other with R12-R17F15 (model 2) buried toward the MRGPRX1 ligand pocket (Figs. 3b–d and Supplementary Fig. 9a–c). Compared with mode 2 the CNF-Tx2 in mode 1 fits better with EM density. We then performed a molecular dynamics simulation by including side chain atoms that were not defined by EM density and the result indicated that model 1 was more stable (Fig. 3e and Supplementary Fig. 9d–g). Consistent with this model, the mutations F15A, R17A, I18A, and the C-terminal truncation (deleting the C-terminal VRI motif) of CNF-Tx2 each significantly decreased CNF-Tx2-induced

MRGPRX1 activities. (Fig. 3d and Supplementary Fig. 9a–b). Notably, R17 of CNF-Tx2 in our simulated structure derived from mode 1 formed Hydrogen bonds or charge interactions with E157[4.60] and D177[5.36]. Compared with mode 1, the CNF-Tx2 in mode 2 lost specific interactions with E157[4.60] and D177[5.36] and formed new contact with F239[6.58]. Importantly, whereas F239A mutation showed no significant effects on CNF-Tx2-induced MRGPRX1 activation, mutations of E157A, and D177A each significantly reduced or totally abolished CNF-Tx2-induced MRGPRX1 activation. Therefore, both MD simulation and mutational analysis indicated that mode 1 of CNF-Tx2 is more favored. We therefore mainly used mode 1 of CNF-Tx2 for further structural analysis. Paralleling the experimentally determined structure, we have used Colabfold[18,19] to predict the binding modes of CNF-Tx2 and BAM8-22 to MRGPRX1. We found that the ligand-binding poses predicted by Colabfold were quite different from the binding patterns of the ligands in our resolved structures (Supplementary Fig. 10a–c). We, therefore, speculated that the experimental data is still needed for analyzing the interaction between peptide ligand and their corresponding receptors.

In our proposed binding mode of CNF-Tx2 in the MD-CNF-Tx2-MRGPRX1 complex structure, the core sequence of R17I18 occupied a similar position to that of the core sequence of R20Y21 of BAM8-22, sitting in a hydrophobic pocket surrounded by Y99[3.29], L160[4.63,] and L240[6.59]. R17 of CNF-Tx2 participated in charge–charge interactions and H-bond networks with E157[4.60] (Fig. 3f; Supplementary Tables 7 and 8). The mutations Y99A, E157A, L160A and L240A significantly damaged CNF-Tx2-induced MRGPRX1 activation (Fig. 3g and Supplementary Fig. 10d–f). Notably, the interactions mediated by Y99[3.29] and E157[4.60] were shared by both BAM8-22 and CNF-Tx2 (Fig. 3h). These results collectively indicated that motifs located in TM3 and TM4 of MRGPRX1, including the residues Y99[3.29]-E157[4.60]-D177[5.36], formed a hydrophobic pocket with an acidic bottom and served as one key determinant for ligand recognition by MRGPRX1.

By comparing the structures of BAM8-22-MRGPRX1-Gq/Gi and CNF-Tx2-MRGPRX1-Gi, as well as the mutational analysis, we were able to identify that a hydrophobic pocket surrounded by Y99[3.29], L160[4.63], and L240[6.59] of MRGPRX1 played an important role in recognition of both C-terminal Y21 of BAM8-22 and I18 of CNF-Tx2. Moreover, the E157[4.60] played central roles in the recognition of C-terminal R17 of CNF-Tx2 and R20Y21 of BAM8-22. In addition to providing structural knowledge for recognition previous proposed C-terminal Rφ motif (φ indicated a hydrophobic residue), we also found that the N-terminal to the Rφ motif, the F15 of CNF-Tx2 is surrounded by large hydrophobic residues of Y82[2.60], Y99[3.29], F236[6.55,] and H254[7.35]. Similarly, the Y17 of BAM8-22 is surrounded by large hydrophobic residues of F236[6.55], F250[7.31,] and H254[7.35] of MRGPRX1. Therefore, we proposed that a C-terminal motif of φ[B17](X[1-2]) R [B20] φ[B21] in the peptide ligand is more preferred by MRGPRX1 (amino acid position of the peptide sequence is

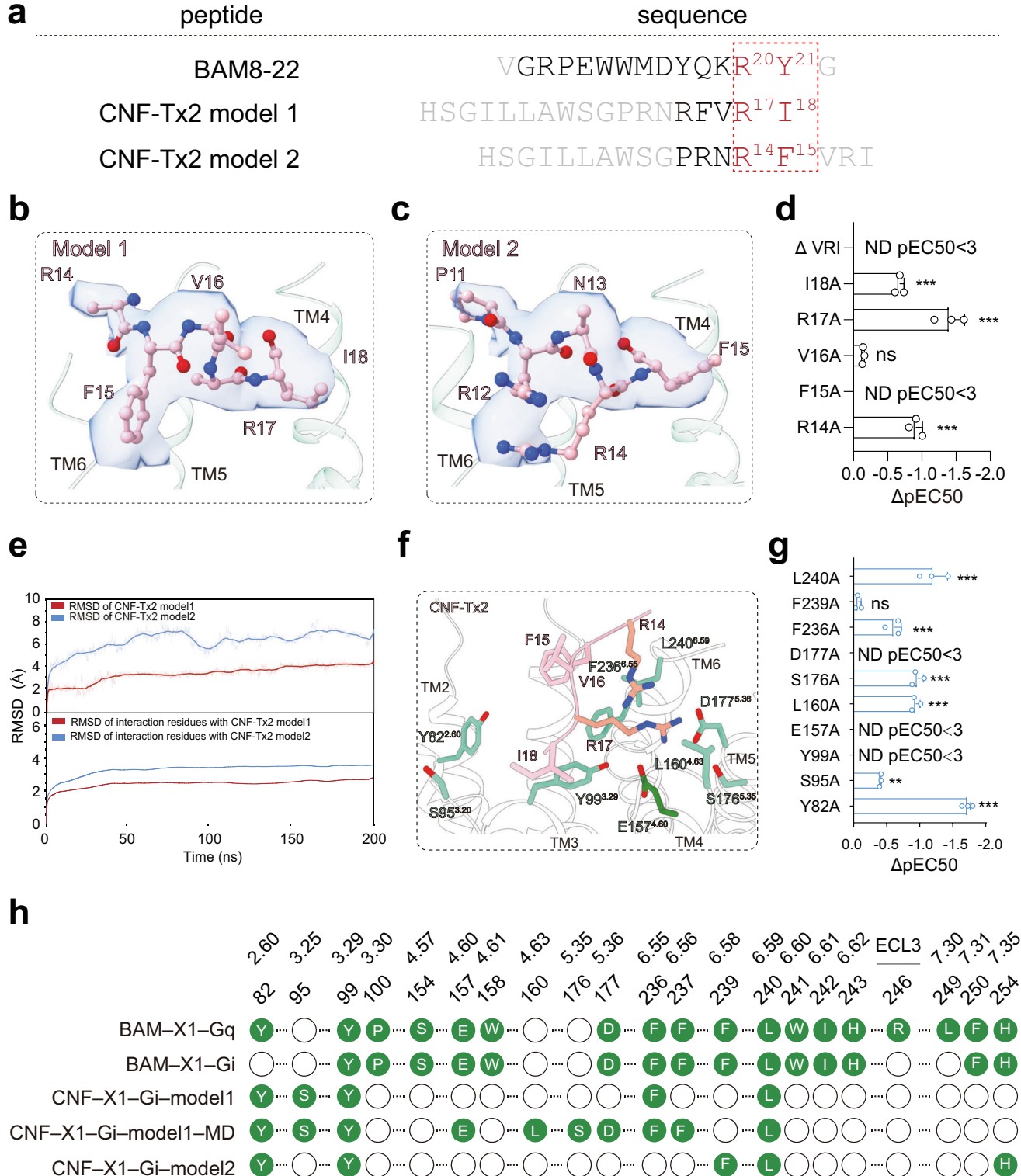

named according to positions in BAM8-22 peptide) (Supplementary Fig. 11a, b).

Consistent with these speculations, we have measured the activities of MRGPRX1 toward γ1-MSH[15], hemoglobin β-chain[20], P60 (part of C5orf29)[21], which showed reasonable potency and efficacy, as previously reported (Supplementary Fig. 11c). Mutations of key motif residues in the $\varphi^{B17}(X_{1-2}) R^{B20} \varphi^{B21}$ motif, significantly weakened the activation of MRGPRX1 by these peptides (Supplementary Fig. 11d–e).

## The active state of MRGPRX1

The active states of MRGPRX1 bound by both Gq and Gi showed several common features conserved in the MRGPRX itch receptor family but distinct from those of other GPCRs. For example, the upper region of TM6 was distorted inward by 3.6 Å at the upper rim (Fig. 4a). This kink was centered on a conserved P231[6.50], stabilized by the conserved hydrogen bond network formation between the phenolic oxygen of Y106[3.36], the main chain carbonyl of G229[6.48] and the main chain amine of G233[6.52], as well as the hydrophobic chain packing of Y106[3.36],

**Fig. 3 | The CNF-Tx2-binding pocket in MRGPRX1-Gi1 complex. a** Peptide ligand sequence of MRGPRX1. Amino acids fitting to EM density were shown in black and the conserved RF/Y-G or RF/Y-amide motif was highlighted in red. **b, c** Structural representation and EM density of CNF-Tx2 model 1 (**b**) and CNF-Tx2 model 2 (**c**). **d** Effects of different CNF-Tx2 mutations on CNF-Tx2 induced Gαi-Gγ dissociation. A bar graph for EC50 was presented. Statistical differences between CNF-Tx2 WT and mutations were determined by two-sided one-way ANOVA with the Tukey test. *$P < 0.05$; **$P < 0.01$; ***$P < 0.001$; ns, no significant difference. ($P =$ ND, 0.002, <0.001, 0.4777, <0.001 from top to bottom). Data from three independent experiments are presented as the mean ± SEM ($n = 3$). **e** The average RMSD value of CNF-Tx2 model1 (red) and CNF-Tx2 model2 (blue) (upper panel) and RMSD of key residues in MRGPRX1 which directly interact with CNF-Tx2 model1 (red) and CNF-Tx2 model2 (blue) (lower panel) during triplicate 200 ns MD simulations. **f** Three-dimensional (3D) representation of the detailed interactions between CNF-Tx2 and

MRGPRX1 in MRGPRX1-Gi1 complex, the simulated residues in CNF-Tx2 and MRGPRX1 were shown in light salmon and forest green. **g** Effects of different mutations within the ligand-binding pocket of MRGPRX1 on CNF-Tx2 induced Gαi-Gγ dissociation. Statistical differences between MRGPRX1 WT and mutations were determined by two-sided one-way ANOVA with Tukey test. *$P < 0.05$; **$P < 0.01$; ***$P < 0.001$; ns, no significant difference. ($P =$ <0.001, 0.9446, <0.001, ND, <0.001, <0.001, ND, ND, 0.0012, <0.001 from top to bottom). Data from three independent experiments are presented as the mean ± SEM ($n = 3$). **h** Barcode representations of interaction patterns of the ligands in the pocket of the MRGPRX1-Gi1 bound by BAM8-22, CNF-Tx2, and MRGPRX1-Gq bound by BAM8-22. Residues of MRGPRX1 that interact with ligands were indicated by the green circle. Residues of MRGPRX2 showing no interaction with BAM8-22 or CNF-Tx2 were indicated by the blank circle.

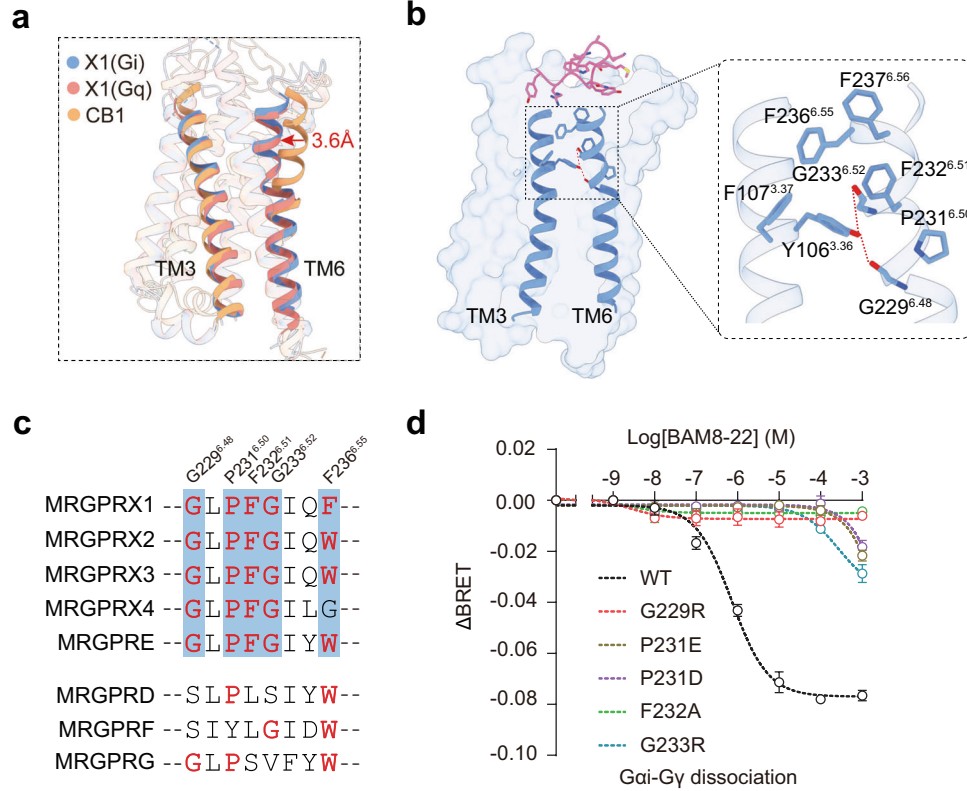

**Fig. 4 | The activation mechanism of MRGPRX1. a** Structural comparison of TM3 and TM6 in the MRGPRX1-Gi1/Gq complexes (cornflower blue/salmon) and CB1-Gi1 complex (orange, PDB: 6N4B). The upper regions of TM6 were twisted inward in MRGPRX1-Gi1/Gq complexes compared with another prototype class A GPCR and indicated a red arrow. **b** Structural representation of the BAM8-22 (hot pink) induced MRGPRX1 activation via TM6 kink. Hydrogen bonds were depicted as red dashed lines. **c** Sequence alignment of the kink position of MRGPRX1 residues with other MRGPR family receptors. The key residues are highlighted in red. **d** Effects of mutations of the residues that were involved in kink formation at TM6 on BAM8-22 induced MRGPRX1 activity, assessed using a Gαi-Gγ dissociation assay. Data are mean ± SEM of three independent experiments for wild-type MRGPRX1 or mutants ($n = 3$).

$F107^{3.37}$, $F232^{6.51}$, $F236^{6.55}$ and $F237^{6.56}$ (Fig. 4b). $F236^{6.55}$ and $F237^{6.56}$ form direct interactions with BAM8-22 in both BAM8-22-MRGPRX1-Gi and BAM8-22-MRGPRX1-Gq models and thus may participate in MRGPRX1 activation by directly sensing the ligands (Fig. 4b). Consistent with this hypothesis, the mutations F236A and F237A each significantly diminished the MRGPRX1 activation induced by BAM8-22 (Supplementary Fig. 8d). Moreover, residues of the key kink motif $G^{6.48}XP^{6.50}F^{6.51}$ $G^{6.52}X_{(2)}F/W^{6.55}$ are conserved across MRGPRX families (Fig. 4c). The mutations $G^{6.48}R$, $P^{6.50}D/E$, $G^{6.52}R$, and $F^{6.51}A$ each impaired MRGPRX1 activation, indicating that the formation of the TM6 upper kink and these conserved key residues play important roles in MRGPRX1 ligand recognition and activation (Fig. 4d).

In addition to the unique kink motif present in the MRGPRX family, MRGPRX1 displayed other known features of active class A

GPCRs, including cytoplasmic region separation between TM3 and TM6 and the $E/D^{3.49}R^{3.50}Y/C^{3.51}$ (E119$^{3.49}$, R120$^{3.50}$, C121$^{3.51}$) and $N^{7.49}P^{7.50}xxY^{7.53}$ (N268$^{7.49}$, P269$^{7.5}$, Y272$^{7.53}$) motifs.

## Coupling of MRPGRX1 with Gi and Gq

Both Gi1- and Gq-mediated MRGPRX1 signaling may be involved in itch sensation, as suggested by studies using a mouse model[8,9]. We, therefore, measured the bias property of several MRGPRX1 ligands for their Gi.vs. Gq activities. The results indicated that the CQ is a Gq bias ligand, and the CNF-Tx2 showed Gi bias when we compared Gq activation over Gi using BAM8-22 as a reference (Supplementary Fig. 11f–h). Further, the structures of the MRGPRX1-Gi1/Gq complexes allowed us to investigate both the Gi1 and Gq coupling mechanisms of MRGRPX1 (Fig. 1a and Fig. 5a–c). A total of eighteen

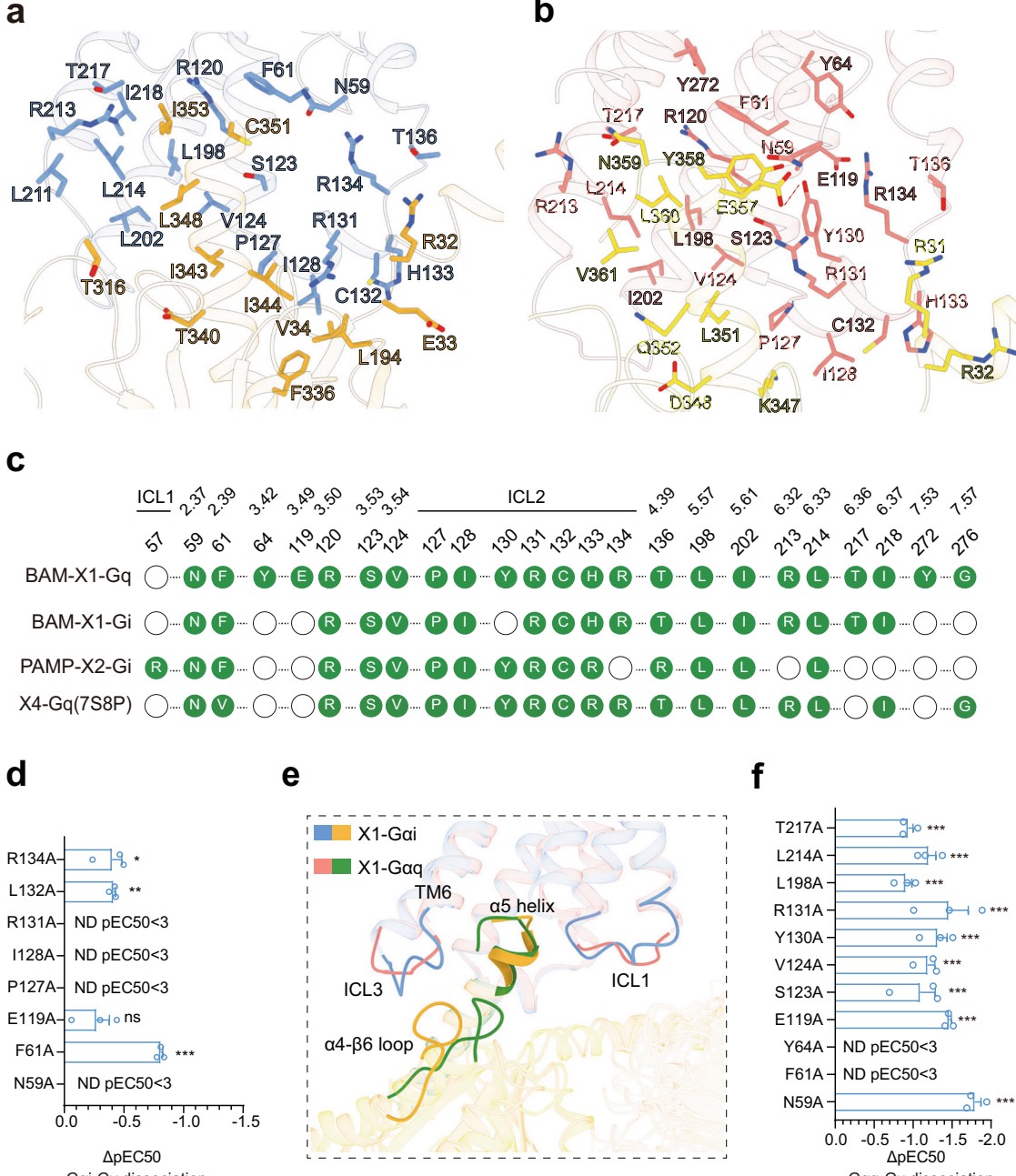

**Fig. 5 | The coupling of MRPGRX1 with Gi1 and Gq. a** Three-dimensional (3D) representation of the detailed interactions between MRGPRX1 and the α5-helix of Gαi. **b** Three-dimensional (3D) representation of the detailed interactions between MRGPRX1 and the α5-helix of Gαq. **c** Comparison of the Gi1/Gq coupling interfaces in cryo-EM structures of BAM8-22-MRGPRX1-Gq, BAM8-22-MRGPRX1-Gi1 and PAMP-12-MRGPRX2-Gi1 complexes. Residues of MRGPRX1 in contact with Gi1/Gq were illustrated as green dots. **d** Effects of mutations in the MRGPRX1 along the Gi trimer interface on BAM8-22 induced Gαi-Gγ dissociation. Statistical differences between WT and mutations were determined by two-sided one-way ANOVA with the Tukey test. *$P < 0.05$; **$P < 0.01$; ***$P < 0.001$, ns, no significant difference. ($P = 0.0109, 0.0091$, ND, ND, ND, $0.1196$, <0.001, ND from top to bottom). Data

from three independent experiments are presented as the mean ± SEM ($n = 3$). **e** The structural representation and comparison of the interfaces between the MRGPRX1-Gi and MRGPRX1-Gq complexes. Ribbon representation: MRGPRX1 bound to Gi is shown in cornflower blue, MRGPRX1 bound to Gq is shown in salmon, Gi is shown in dark orange, Gq is shown in green. **f** Effects of mutations in the MRGPRX1 along the Gαq trimer interface on BAM8-22 induced Gαq-Gγ protein dissociation. Statistical differences between WT and mutations were determined by two-sided one-way ANOVA with the Tukey test. *$P < 0.05$; **$P < 0.01$; ***$P < 0.001$, ns, no significant difference. ($P = 0.0003$, <0.001, <0.0004, <0.001, <0.001, <0.001, <0.001, $P < 0.001$, ND, ND, <0.0001 from top to bottom) Data from three independent experiments are presented as the mean ± SEM ($n = 3$).

residues of TM2, TM3, TM4, TM5, TM6, and ICL2 of MRGPRX1 were observed to form direct interactions with both the Gi1 trimer and Gq trimer (Fig. 5a–c). More MRGPRX1-Gq contacts were found than in the MRGPRX1-Gi1 complex, including additional contacts mediated by TM7, as well as three more contact residues in TM3 and ICL2 of MRGPRX1.

The two successive hydrophobic residues close to the α5-helix end of Gαi1, I344[G.H5.16,] and L348[G.H5.20,] formed hydrophobic packing with V124[3.54,] P127[3.57,] and I202[5.61] at the intracellular ends of the TM bundles of MRGPRX1 (Fig. 5a and Supplementary Fig. 12a). Notably, I128[ICL2] of MRGPRX1 sat in a hydrophobic pocket created by L194[G.S3.01,] F336[G.H5.08,] T340[G.H5.12,] I343[G.H5.15,] R131[ICL2,] and C132[ICL2] of MRGPRX1

formed polar or hydrophobic interactions with E33$^{G.S1.01}$ and V34$^{G.S1.02}$ of Gαi (Fig. 5a; Supplementary Fig. 12b and Supplementary Table 9). The observed Gi1 interface of BAM8-22-MRGPRX1 was verified by mutagenesis (Fig. 5d; and Supplementary Figs. 12c, d and 13).

Compared with that of Gαi, the structure alignment indicated that the α5-helix of Gαq tilted more toward TM6 and ICL3 of MRGPRX1 (Fig. 5e). In particular, the bulky end of the α5-helix of Gαq, including N359$^{G.H5.24}$, L360$^{G.H5.25}$, and V361$^{G.H5.26}$, formed extensive contacts with V124$^{3.54}$, L198$^{5.57}$, R213$^{6.32}$, L214$^{6.33}$ and T217$^{6.36}$, which are constituted by both hydrophobic and polar interactions (Fig. 5b; Supplementary Figs. 12e and 13). N-terminal to the bulky end of the α5-helix, Y358$^{G.H5.23}$ of Gαq formed hydrophobic packing with Y64$^{2.42}$, F61$^{2.39}$, and Y130$^{ICL2}$ of MRGPRX1. E357$^{G.H5.22}$ of Gαq not only formed a salt bridge with the R131$^{ICL2}$ of MRGPRX1 but also constituted a polar network together with Y358$^{G.H5.23}$ of Gαq and Y130$^{ICL2}$ of MRGPRX1 (Fig. 5b; Supplementary Figs. 12f, 13 and Supplementary Table 10). Consistent with these observations, mutations of these residues significantly impaired BAM8-22-induced Gαq activation downstream of MRGPRX1 (Fig. 5f and Supplementary Figs. 12g, h and 13). Collectively, whereas conserved interactions in TM3 and ICL2 accounted for both Gαi and Gαq coupling in response to MRGPRX1 activation, the interaction between ICL1, the cytoplasmic end of TM6 of MRGPRX1, and the bulky end of the α5-helix of Gαq contributed to most of the specific Gαq coupling mechanisms. In addition to our solved BAM8-22-MRGPRX1-Gq complex structure, recent studies also reported structures of both MRGPRX1 and MRGPX4 in complex with Gq.[16,22] By comparison, our BAM8-22-MRGPRX1-Gq complex structure did not show significant difference compared with recently published BAM8-22-MRGPRX1-Gq structure (PDB ID: 8DWC), despite that we used different Gq chimera constructs (Supplementary Fig. 14). However, MRGPRX1-Gq structure showed a significantly difference compared to MRGPRX4-Gq structure (PDB ID: 7S8P) (Supplementary Fig. 15).

## Discussion

The cryo-EM structures of MRGPRX1 in complex with the endogenous ligands BAM8-22 and CNF-Tx2 and the detailed pharmacological characterization in this study revealed that motifs located in TM3 and TM4 of MRGPRX1, including residues Y99$^{3.29}$/P100$^{3.30}$-E157$^{4.60}$/W158$^{4.61}$, formed a hydrophobic pocket with an acidic bottom and served as a key determinant for the recognition of the "RF/RY" motif of the peptide ligands of MRGPRX1. Similar to the recently solved structures of MRGPRX2 and MRGPRX4 in complex with their peptic or chemical ligands, MRGPRX1 has a shallower ligand pocket than many other class A GPCRs. This specific feature of the shallower orthosteric ligand pocket, as well as the conformational plasticity of the ECL loops of both MRGPRX1 and MRGPRX2 in the recognition of different ligands, may be key to the mechanisms of itch sensation by accommodating multiple different itch substances.

Notably, the peptide ligand-binding orientation of MRGPRX1 is distinct from that of MRGPRX2 with its corresponding ligands, as illustrated by our previous studies: the peptide ligand anchors to the extracellular end of TM3 and acidic residue D/E$^{4.60}$ via the N-terminal part in MRGPRX2 but the C-terminal part in MRGPRX1. This different binding mode could help explain the exclusive activities of several peptidic allergens/itchy substances toward these two MRGPRX members. Moreover, whereas MRGPRX2 may favor a peptide containing a motif φ$^{p9}$(X$_{0-1}$) R/K$^{p10}$(X$_2$) φ$^{p13}$(X$_{2-3}$) φ$^{p16}$(X$_3$) R/K$^{p20}$, here we proposed that MRGPRX1 potentially recognize a peptide containing C-terminal motif of φ$^{B17}$(X$_{1-2}$) R$^{B20}$ φ$^{B21}$ motif. Interestingly, below the ligand-binding pocket, the key motifs G$^{6.48}$XP$^{6.50}$F$^{6.51}$G$^{6.52}$X$_{(2)}$F/W$^{6.55}$ are conserved across MRGPRX families. These motifs are structurally essential for the inward tilting of the upper region of TM6 and play key roles in MRGPRX1 activation. By comparing BAM8-22-MRGPRX1 in complex with both Gi1 and Gq proteins, we found that M[15] and Y[17] of BAM8-22 assume different poses and form different interactions with the

residues F236$^{6.55}$, R246$^{ECL3}$, and L249$^{7.30}$ of MRGPRX1. On the cytoplasmic side, TM3 and ICL2 of MRGPRX1 form specific interactions with the bulky end of the α5-helix of Gαq. These observed structural features offer preliminary insight into the signal transduction of MRGPRX1 toward Gαi and Gαq effectors. Collectively, our studies provide important structural and pharmacological insights into the understanding of the itch sensation, activation, and G-protein signaling downstream of MRGPRX1.

## Methods

### Construct

The human MRGPRX1 gene was cloned into pcDNA3.1 and pFastBac1 vectors for functional assays and protein expression, respectively. To facilitate expression and purification, a BRIL epitope was inserted into the N-terminal of wild-type MRGPRX1. The engineered Gαq construct with two mutations (G203A and A326S) was generated on the basis of a mini-Gαs/Gq$_{71}$ scaffold in which the N-terminus was replaced by corresponding sequences of Gαi1 to facilitate the binding of scFv16[23]. The human Gβ1 that harbors the N-terminal 6 × His and the human Gγ2 were cloned into the pFastbacdual vector. All of the MRGPRX1 mutations were generated using the Quikchange mutagenesis kit (Stratagene). All of the constructs were verified by DNA sequencing.

### Protein expression

*Spodoptera frugiperda* (Sf9) cells were purchased from Expression Systems (Cat 94-001S) and were grown in ESF 921 medium to a density of $2.5 \times 10^6$ cell/ml and then were co-infected with four separate baculoviruses (Bac-to-Bac Baculovirus system, Invitrogen) at a ratio of 1:2:1:1 for MRGPRX1, Gi1/Gq, Gβ1γ2, and scFv16, respectively. After 48 h culture, the cells were collected by centrifugation, and cell pellets were stored at −80 °C.

### Complex formation and purification

The supernatant was collected by centrifugation at $65,000 \times g$ for 30 min, and the solubilized complex was purified by anti-FLAG affinity resin. The resin was washed with 20 column volumes of 20 mM HEPES, pH 7.4, 100 mM NaCl, 2 mM MgCl$_2$, 5 mM CaCl$_2$, 100 μM BAM8-22 (or CNF-Tx2), 0.01% (w/v) LMNG and 0.001% (w/v) CHS. The complex was then eluted in buffer containing 20 mM HEPES, pH 7.4, 100 mM NaCl, 2 mM MgCl$_2$, 100 μM BAM8-22 (or CNF-Tx2), 0.01% (w/v) LMNG, 0.001% (w/v) CHS, 5 mM EGTA and 0.2 mg/ml FLAG peptide. After concentration using an Amicon Ultra Centrifugal Filter (MWCO, 100 kDa), the MRGPRX1-Gi1 complex was subjected to size-exclusion chromatography on a Superdex 200 Increase 10/300 column (GE Healthcare). The complex fractions were collected and concentrated individually for EM experiments.

### Cryo-EM data acquisition

The purified BAM8-22-MRGPRX1-Gi1 complex (3.0 μl) at 5.0 mg/ml, BAM8-22-MRGPRX1-Gq complex (3.0 μl) at 4.0 mg/ml, and the CNF-Tx2-MRGPRX1-Gi1 complex (3.0 μl) at 4.5 mg/ml was applied onto a glow-discharged holey carbon grid (Quantifoil R1.2/1.3), and subsequently vitrified using a FEI Vitrobot Mark IV (Thermo Fisher Scientific). The cryo-grids were initially screened at a nominal magnification of ×92,000 in an FEI Talos Arctica microscope (200 kV), equipped with an FEI Ceta camera. High-quality grids were transferred to an FEI Titan Krios electron microscope (Thermo Fisher Scientific) equipped with a Gatan K2 or K3 Summit direct electron detector and a Gatan Quantum-LS Energy Filter (GIF, slit width of 20 eV).

For the BAM8-22-MRGPRX1-Gq complex dataset, 5601 movies were collected on a Titan Krios equipped with a Gatan K3 direct electron detection device at 300 kV with a magnification of 81,000, corresponding to a pixel size 1.04 Å. We collected a total of 36 frames accumulating to a total dose of 50 e$^-$/Å$^2$ over 2.5 s exposure on each TIF format movie.

For the CNF-Tx2-MRGPRX1-Gi1 complexes, 3085 movies were collected on a Titan Krios equipped with a Gatan K2 direct electron detection device at 300 kV with a magnification of 130,000, corresponding to a pixel size 1.08 Å. The total exposure time was 8 s, resulting in an accumulated dose of 50 electrons per $Å^2$ and a total of 32 frames per movie.

For the BAM8-22-MRGPRX1-Gi1 complexes, all 5540 movies were collected on a Titan Krios equipped with a Gatan K3 direct electron detection device at 300 kV with a magnification of 130,000, corresponding to a pixel size 0.89 Å. The total exposure time was 3 s, resulting in an accumulated dose of 60 electrons per $Å^2$ and a total of 32 frames per movie.

## Image processing and 3D reconstruction
Original image stacks were summed and corrected for drift and beam-induced motion at the micrograph level using the MOTIONCOR2[24]. Images without dose weighting were used to determine the parameters of the contrast transfer function by CTFFIND. All 2D and 3D classification and refinement were performed with RELION 4.0. The local resolution map was generated using ResMap[25].

For the BAM8-22-MRGPRX1-Gi1 sample, a total of 5540 usable micrographs were collected and 3,628,139 particles were picked for a cascade of 2D and 3D classification with a binning factor of two. About 50% of particles were removed during several rounds of 2D and 3D classification, and the good particles were split into four classes during the final round of 3D classification. After the final round of 3D classification, one class displayed fine structural details that were subjected to high-resolution refinement (without binning), resulting in global maps, state (925,644 particles) at resolutions of 3.0 Å map (gold-standard FSC 0.143).

For the BAM8-22-MRGPRX1-Gq sample, a total of 5601 qualified micrographs were collected and 11,127,531 particles were picked for a cascade of 2D and 3D classification with a binning factor of two. About 55% of particles were removed during several rounds of 2D and 3D classification, and the good particles were split into six classes during the final round of 3D classification. After the final round of 3D classification, three classes showed detailed features for all subunits were obtained. A final class with 1,316,443 particles was subjected to high-resolution refinement (without binning), resulting in a 2.7 Å map (gold-standard FSC 0.143).

For the CNF-Tx2-MRGPRX1-Gi1 sample, a total of 3085 usable micrographs were collected and 1,468,249 particles were picked for a cascade of 2D and 3D classification with a binning factor of two. About 45% of particles were removed during several rounds of 2D and 3D classification, and the good particles were split into six classes during the final round of 3D classification. After the final round of 3D classification, one stable class containing 315,448 particles showed detailed features for all subunits, which was subjected to high-resolution refinement (without binning), resulting in a 2.8 Å map (gold-standard FSC 0.143).

## Model building and refinement
For the structure of BAM8-22-MRGPRX1-Gi1, CNF-Tx2-MRGPRX1-Gi1, BAM8-22-MRGPRX1-Gq complexes, the initial homology model was generated using SWISS-MODEL website. Each subunit of MRGPRX1-Gi1 was manually docked into the EM density map with UCSF Chimera[26]. Subsequent model adjustment and rebuilding were done with Coot[27]. The ligands of BAM8-22 and CNF-Tx2 were both built ab initio in Coot. Models were further refined against the cryo-EM density maps using Phenix.real_space_refinement with geometry restraints and secondary structures restraints imposed.

## Molecular dynamics simulations
The initial CNF-Tx2-MRGPRX1 complex model used for molecular dynamics simulation was the cryo-EM structural model of the CNF-Tx2-bound MRGPRX1 receptor part extracted from CNF-Tx2-MRGPRX1-Gi1 complex. The 200 ns molecular dynamics simulations were carried out by GROMACS2019.5 with CHARMM36m force field[28]. All the input files were generated by CHARMM-GUI[29]. The structure of the CNF-Tx2 model1-MRGPRX1 and CNF-Tx2 model2-MRGPRX1 complexes were embedded into a pre-equilibrated and periodic structure of 1-palmitoyl-2-oleoyl-sn-glycero-3-phosphatidylcholine membrane, 0.15 M NaCl was added to balance the charge of the system using the Monte Carlo method. Next, the systems of CNF-Tx2 model1-MRGPRX1 and CNF-Tx2 model2-MRGPRX1 complexes were solvated into periodic TIP3P model water box with a size of roughly $70 \times 70 \times 100$ $Å^3$ by replacement methods using the membrane orientation calculated by the Orientations of Proteins in Membranes database. The box type was set as hexagonal. The systems of hexagonal were first subjected to energy minimization for 10,000 steps, of which the first 5,000 steps were performed using the steepest descent method and the remaining 5000 steps using the conjugated gradient method. The systems were heated from 0 to 310 K in the NVT ensemble for 1000 ps. Following this, production simulations were run at 1 atm in the NPT ensemble for 1000 ps with 10.0 kcal $mol^{-1} Å^{-2}$ harmonic restraints. The particle mesh Ewald method was used for CNF-Tx2 model1-MRGPRX1 and CNF-Tx2 model2-MRGPRX1 complexes to calculate electrostatic interactions with a cutoff of 12 Å. The bonds involving hydrogen were kept fixed using the SHAKE algorithm during each integration time step of 2 fs. Each simulation condition was repeated three times. (Supplementary Fig. 9d–g)

## Gαi-Gγ and Gαq-Gγ dissociation assay
Human Embryonic Kidney 293 (HEK293) cells were obtained from the Cell Resource Center of Shanghai Institute for Biological Sciences (Chinese Academy of Sciences, Shanghai, China) and cultured in DMEM with 10% FBS for functional studies. Transient transfection in the current study was performed with Lipofectamine 2000 (Invitrogen), according to the manufacturer's instructions. For studying the peptide-induced G-protein activation through MRGPRX1, G-protein BRET probes were generated according to previous publications[13,30,31]. HEK293 cells were transfected with plasmids MRGPRX1 and G-protein probes. After 24 h, cells were reseeded into a 96-well plate with a density of 30,000-50,000 cells per well and incubated for an additional 24 h at 37 °C. For peptide-stimulated activity measurement, the BRET signal between Rluc8 and GFP2 was recorded after the addition of peptide ligands (final concentration:$10^{-10} – 10^{-3}$ M) and coelenterazine 400a. The BRET signal was calculated as the ratio of the GFP2 emission to the Rluc8 emission.

## Enzyme-linked immunosorbent assay
The MRGPRX1 plasmids transfected cells were seeded at a density of $5 \times 10^4$ cells per well into 96-well plates and cultured in a 37 °C incubator for 24 h. Cells were fixed by polyformaldehyde for 10 min and then blocked with 5% (w/v) BSA for at least 1 h at room temperature. The washed ELISA plates were probed with an anti-FLAG antibody (Sigma Aldrich, Catalog # F1804, 1:1000) and then incubated with a secondary anti-mouse antibody (Thermo Fisher, Cat# 31430, 1:5000), supplementing with 100 µl TMB substrate (Millipore) until color turned blue. The reaction was stopped with an equal amount of 0.25 M HCl and analyzed by a microplate reader at 450 nm wavelength. The expression levels of the wild-type MRGPRX1 or selective mutants were iteratively adjusted by the quantity of transfected plasmids to assure equal receptor expression in the BRET assay.

## Reporting summary
Further information on research design is available in the Nature Portfolio Reporting Summary linked to this article.

## Data availability

The data that support this study are available from the corresponding authors upon request. The cryo-EM maps have been deposited in the Electron Microscopy Data Bank (EMDB) under accession code EMD-36232 (BAM8-22- MRGPRX1-Gq), EMD-36229 (CNF-Tx2- MRGPRX1-Gi), and EMD-36233 (BAM8-22- MRGPRX1-Gi). The coordinates have been in the Protein Data Bank (PDB) under accession codes 8JGF (BAM8-22-MRGPRX1-Gq), 8JGB (CNF-Tx2- MRGPRX1-Gi), and 8JGG (BAM8-22-MRGPRX1-Gi). The source data underlying Figs. 1e, 2c, d, 2g, 3d, 3g, 4d, 5d, 5f, and Supplementary Figures 8a, 8c, d, 9a, b, 9g–i, 10c, 10f, g, 11c, d, 11g, h are provided as a Source Data file. Source data are provided with this paper.

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

## Acknowledgements

We acknowledge support from the National Key R&D Program of China (2019YFA0904200 to J.-P.S., 2022YFC2702603 to F.Y.), the National Science Fund for Distinguished Young Scholars Grant (81825022 to J.-P.S.), the National Science Fund for Excellent Young Scholars (82122070 to F.Y.), the National Natural Science Foundation of China (91939301 to J.-P.S.; 92057121 to X. Yu; 92168120 to L.T.), the Major Basic Research Project of Shandong Natural Science Foundation (ZR2020ZD39 to J.-P.S.), the Beijing Natural Science Foundation (Z200019 to J.-P.S. and L.T.). This work was partially supported by the National Natural Science Foundation (32171187 to Y.J., 82121005 to Y.J. and H.E.X., 32130022 to H.E.X.), the Ministry of Science and Technology (China) grants (2018YFA0507002 to H.E.X.), the Shanghai Municipal Science and Technology Major Project (2019SHZDZX02 to H.E.X.), Shanghai Municipal Science and Technology Major Project (H.E.X.); the CAS Strategic Priority Research Program (XDB37030103 to H.E.X.), and Key Tasks of LG Laboratory (LG202101-01-03 to H.E.X.). We thank the staff at the Department of Biochemistry and Molecular Biology, Shandong University School of Medicine; the staff at the Shanghai Advanced Center for Electron Microscopy, Shanghai Institute of Material Medica; the staff at Translational Medicine Core Facility of Advanced Medical Research Institute, Shandong University, and the staff at the cryo-EM facilities of Southern University of Science and Technology.

## Author contributions

J.-P.S., H.E.X., and F.Y. organized the whole project; H.E.X., J.-P.S., and J.-Y.Z. guided all of the structural analysis; J.-P.S., F.Y., L.T. and J.-Y.Z. designed the Gi-Gq dissociation assay; L.-L.G., G.-X.F., L.T., Y.-M. Zhuang. generated the MRGPRX1 insect cell expression construct, established the BAM8-22/CNF-Tx2-MRGPRX1-Gi1-scFv16 complexes purification protocol, and prepared samples for the cryo-EM. Y.-M.

Zhang., and C.-Z.Y. established the BAM8-22-MRGPRX1-Gq-scFv16 complex purification protocol and prepared samples for the cryo-EM; G.-X.F., L.T., Z.-Y.L., and Y.M. Zhuang. prepared the cryo-EM grids, collected the cryo-EM data of the BAM8-22/CNF-Tx2-MRGPRX1-Gi1-scFv16; Y.-M. Zhang. and P.-Y.X. collected the cryo-EM data of the BAM8-22-MRGPRX1-Gq-scFv16 complex; C.-Y.X. and Z.-M.L. performed the cryo-EM map calculation of the BAM8-22/CNF-Tx2-MRGPRX1-Gi1-scFv16 complexes. H.E.X., Y.M. Zhang, and Q.-N.Y. performed the cryo-EM map calculation of the BAM8-22-MRGPRX1-Gq-scFv16; L.-L.G., G.-X.F., Q.L., and L.T. performed the model building and refinement of the BAM8-22/CNF-Tx2-MRGPRX1-Gi1-scFv16 complexes; Y.-M. Zhang. performed the model building and refinement of the BAM8-22-MRGPRX1-Gq-scFv16; F.Y., X.Y., and J.-Y.Z. designed all of the mutants for the ligand-binding pocket; L.T., Y.-M. Zhuang., L.L., Y.L., N.-K.R., C.-Z.W. generated all of the MRGPRX1 constructs and mutants for the cell-based G-protein activity assays; L.-L.G., G.-X.F., L.T., Q.L., and Y.-M. Zhuang performed Gαi -Gαq dissociation assay; J.-P.S., F.Y., L.T., D.-W.K., and J.-Y.Z. designed the peptide mutation scanning assay, and designed the mutagenesis for each residue in BAM8-22; C.Z., M.-H.Z., K.-K.Z., and S.-X.P. performed MD simulations; G.-X.F., L.T., Y.-M. Zhuang., X.L., and Cheng Z. performed the peptide mutation scanning assay; J.-P.S. wrote the initial manuscript and H.E.X., Y.J., X.-Z.D., F.Y., and J.-Y.Z. revised the manuscript. G.-X.F. finished the main figure 1b, c, and e; G.-X.F. and L.T. finished the main figure 2c–e, figure 3d, g, figure 4c, d, and figure 5d, e; Y.-M. Zhuang. finished the main figure 2f, g, figure 3a–c, f, h, and figure 4a; C.Z. finished the main figure 4e; L.-L.G. finished the main figure 1a, d, figure 2a, b, figure 4b, and figure 5f; F.Y. finished the main figure 5a–c; L.T. finished the Supplementary figure 1; Y.-M. Zhang and C.-Y.X. finished the Supplementary figures 2–4; Y.-M. Zhuang finished the Supplementary figures 5–7; G.-X.F. finished the Supplementary figures 8–10; L.T. finished the Supplementary fig. 11.

## Competing interests

The authors declare no competing interests.
