## [Peer Review File · Nature Communications]

Ligand recognition and G protein coupling of the human itch receptor MRGPRX1Reviewers' Comments:

Reviewer #1:

Remarks to the Author:

Review of "Ligand recognition and G protein coupling of the human itch receptor MRGPRX1" by Guo et al.

In this manuscript the authors present the first structure of full length MRGPRX1, a G-protein coupled receptor involved in itch perception and pathology. Structures of MRGPRX1 in complex with two different ligands (BAM8-22 and CNF-Tx2) reveal the molecular determinants of ligand binding and help to explain the ligand specificity of this receptor relative to other Mas-related GPCRs. Structural characterization of MRGPRX1 in complex with Gi and Gq serve to further elucidate mechanisms of signal transduction.

This manuscript is a direct follow up to work from a subset of these authors on the structure of MRGPRX2 (reference 13). Related modes of ligand and G protein binding are observed for MRGPRX1 compared to MRGPRX2. The biological and therapeutic implications of characterizing Mas-related GPCR function is high and the structure presented here represents an important and substantial contribution to these efforts. The data presented here are high quality and the conclusions are largely sound (with some technical questions). The work could be suitable for publication in Nature Communications through addressing the comments and questions below.

Major comments:

The authors share some discussion of consensus sequences for predicting binding of ligands to MRGPRX1 (refs 14+15) and how new structural data can rationalize differences in ligand specificity for MRGPRX1 versus MRGPRX2. The manuscript would be more impactful if the authors could more explicitly describe how new structural data informs previously published trends for identifying Mas-related GPCR ligands. Based on this structural data is it possible to predict other naturally occurring ligands that might result in receptor activation? Are there new insights/rules from structural data published here for predicting whether a given ligand binds to MRGPRX1 or MRGPRX2?

The authors use a MRGPRX1 fused at its N terminus to BRIL for structural studies. Could the authors comment on whether this construct behaves similarly to BRIL-free MRGPRX1 in pharmacological assays? This comparison is important for understanding structural data versus pharmacological data.

The signaling properties of many ligands and their mutants (as well as receptor mutants) is described with a single pharmacological parameter (pEC50). This seems to leave out important information about maximal responses (efficacy). Addition of this information would be helpful for evaluating the impact of mutations on signaling.

Receptor signaling through Gi and Gq transducers is evaluated. It seems that some ligands show differential signaling through these pathways. There is a great deal of interest in ligands that show functional selectivity for signaling through different pathways. It would be helpful to see these differences quantified through ligand bias calculations.

For comparing the two different models of CNF-Tx2 interaction with receptor (Figure 3A) it seems that deleting the C terminal VRI motif would allow assessment of the importance of these residues for receptor activation. Is there a reason not to test this directly?

Minor comments:

Many of the ligand residues do not make contact with the receptor directly. Based on the structure activity relationship studies in the manuscript can the authors comment a bit more on the role of ligand residues not directly involved in receptor binding?

The interaction of peptidic ligands with receptors such as GPCRs is an active area of inquiry for newly developed modeling algorithms. Could the authors perform a simple comparison of experimental structures with those produced by a modeling algorithm such as AlphaFold2? Either success or failure of AlphaFold2 in predicting structural details would be informative.

On line 52: "GPCRs" should be "GPCR"

On line 91 the language used is confusing. Saying that GPCRs are known to couple to TRPA1 is not clear. GPCRs typically couple to G proteins and beta-arrestin. Do these GPCRs couple to TRPA1 in an analogous way or is TRPA1 a downstream response?

On line 224: "was" should be deleted

Reviewer #2:

Remarks to the Author:

The manuscript by Guo et al., reports the cryo-EM structures of MRGPRX1-Gi1 in complex with BAM8-22 or CNF-Tx2 and MRGPRX1-Gq in complex with BAM8-22, revealing a unique shallow ligand binding pocket at the extracellular ends of TM3 and TM4 for peptidic allergen recognition. They also describe the conserved kink motif present in the MRGPRX family for MRGPRX1 activation. In addition, they reveal both the Gi1 and Gq coupling mechanisms of MRGPRX1 and found that TM3 and ICL2 of MRGPRX1 form specific interactions with the bulky end of the $\alpha 5$ helix of Gq contributed to most of the specific Gq coupling mechanisms. These observations are nicely verified by their mutagenesis studies.

Overall, the manuscript describes elegant and rigorous structural analysis and biochemical experiments. Their maps look like they are good quality. The mechanism that is proposed is reasonable and is based on well-designed experiments that are suggested by the structure. However, before publication could be recommended, a few, mostly minor, issues should be addressed:

- 1) As far as I know, the structure of Gq-coupled MRGPRX1 with BAM8-22 has been already reported (PDB 8DWC). The authors should compare with their structure and state the differences and similarities.
- 2) Considering the structure of MRGPRX4-Gq complex has been resolved, it would be better if authors could compare their differences and similarities.
- 3) Page 3, line 52: "MGRPRX1" "MRGPRX1"
- 4) Page 4, line 91 and 93 "Mrgprc11" "MrgprC11"
- 5) Page 5, line 107, 121 and 123 : "2.9Å" "2.98Å" "2.8Å" "2.84Å"
- 6) Page 9, line 218: "is fits" "fits"
- 7) Page 9, line 224: There is no figure provided to fit the description of the interactions in mode 2.
- 8) Please check "Cryo-EM data acquisition" again: The pixel size does not fit the pixel size given in Supplementary Table 1, as well as defocus range and total exposure electron. Meanwhile a dose rate of about 7.8 electrons per Å² per second and total exposure time of 8 s should be the parameters of K2 camera other than K3, the authors need clarify the detailed data collection parameters for each structures.
- 9) The resolution showed in Supplementary Fig. S2c is 2.85Å which is not consistent with the labeled 2.7Å.
- 10) In Supplementary Fig. S2d, local resolution of the GPCR part seems unreasonable.
- 11) Fig. 2c, Supplementary Fig. S2a,b : R20K R20A
- 12) The clashscore of three structures in Supplementary Table 1 is not consistent with the validation reports.
- 13) According to the information of 5.3.2 protein sidechains in validation reports, more efforts need to be done to correct outliers. And it would be better to provide formal version of validation reports next time.

Responses to reviewers' comments

We thank the reviewer for their valuable time in reviewing our manuscript and the constructive suggestions that they have provided. We have carefully taken these comments into consideration in preparing a revised version for our manuscript, which has resulted in a more thorough and clear manuscript. Please find below a point-by-point response to the reviewer with our responses in Blue and the reviewers' comments in Red.

Reviewer #1 (Remarks to the Author):

Review of "Ligand recognition and G protein coupling of the human itch receptor MRGPRX1" by Guo et al.

In this manuscript the authors present the first structure of full length MRGPRX1, a G-protein coupled receptor involved in itch perception and pathology. Structures of MRGPRX1 in complex with two different ligands (BAM8-22 and CNF-Tx2) reveal the molecular determinants of ligand binding and help to explain the ligand specificity of this receptor relative to other Mas-related GPCRs. Structural characterization of MRGPRX1 in complex with Gi and Gq serve to further elucidate mechanisms of signal transduction.

Reply: We thank the reviewer for his/her positive comments.

This manuscript is a direct follow up to work from a subset of these authors on the structure of MRGPRX2 (reference 13). Related modes of ligand and G protein binding are observed for MRGPRX1 compared to MRGPRX2. The biological and therapeutic implications of characterizing Mas-related GPCR function is high and the structure presented here represents an important and substantial contribution to these efforts. The data presented here are high quality and the conclusions are largely sound (with some technical questions). The work could be suitable for publication in Nature Communications through addressing the comments and questions below.

Reply: We thank the reviewer for his/her positive comments.

Major comments:

1) The authors share some discussion of consensus sequences for predicting binding of ligands to MRGPRX1 (refs 14+15) and how new structural data can rationalize differences in ligand specificity for MRGPRX1 versus MRGPRX2. The manuscript would be more impactful if the authors could more explicitly describe how new structural data informs previously published trends for identifying Mas-related GPCR ligands. Based on this structural data is it possible to predict other naturally occurring ligands that might result in receptor activation? Are there new insights/rules from structural data published here for predicting whether a given ligand binds to MRGPRX1 or MRGPRX2?

Reply: We thank the reviewer for his/her helpful suggestions. Previous report has suggested that MRGPRX1 was able to sense endogenous and exogenous peptides sharing a conserved sequence of RF/Y-G or RF/Y-amide near their C-terminal. Here, by comparison of the structures of BAM8-22-MRGPRX1-Gq/Gi and CNF-Tx2-MRGPRX1, as well as the mutational analysis, we were able to identify that a hydrophobic pocket surrounded by Y99^{3,29}, L160^{4,63} and L240^{6,59} of MRGPRX1 played an important role in recognition of both C-terminal Y²¹ of BAM8-22 and I¹⁸ of CNF-Tx2. Moreover, the E157^{4,60} played central roles in recognition of C-terminal R¹⁷ of CNF-Tx2 and R²⁰Y²¹ of BAM8-22. In addition to providing structural knowledge for recognition previous proposed C-terminal R ϕ motif (ϕ indicated a hydrophobic residue), we also found that N-terminal to the R ϕ motif, the F¹⁵ of CNF-Tx2 is surrounded by large hydrophobic residues of Y82^{2,60}, Y99^{3,29}, F236^{6,55} and H254^{7,35}. Similarly, the Y¹⁷ of BAM8-22 is surrounded by large hydrophobic residues of F236^{6,55}, F250^{7,31} and H254^{7,35} of MRGPRX1. Therefore, we proposed that a C-terminal motif of $\phi^{B17}(X_{1-2}) R^{B20} \phi^{B21}$ in the peptide ligand is more preferred by MRGPRX1 (amino acid position of peptide sequence is named according positions in BAM8-22 peptide). This motif is distinct from our previously identified peptide motif recognized by MRGPRX2, which is $\phi^{p9}(X_{0-1}) R/K^{p10}(X_2) \phi^{p13}(X_{2-3}) \phi^{p16}(X_3) R/K^{p20}$ (Fig. R1a-R1b and Supplementary Fig. 10a-10b in the revised manuscript). We have incorporated these discussions in the “discussion section” of the revised manuscript.

Consistent with these speculations, we have measured the activities of MRGPRX1 toward γ 1-MSH, hemoglobin β -chain, etc, which showed reasonable potency and efficacy, as

previously reported (Fig. R1c and Supplementary Fig. 10c in the revised manuscript). Mutations of key motif residues in the $\phi^{B17}(X_{1-2}) R^{B20} \phi^{B21}$ motif, significantly weakened the activation of MRGPRX1 by these peptides (Fig. R1d- R1e and Supplementary Fig. 10d-10e in the revised manuscript).

Fig. R1 (also shown in Supplementary Fig.10) Sequence alignment of peptide common motif recognized by MRGPRX1 and MRGPRX2.

(a) Peptide ligand sequences of MRGPRX2. Sequence comparisons of several peptide-based allergens with similarities to the $\phi^{p9}(X_{0-1}) R/K^{p10}(X_2) \phi^{p13}(X_{2-3}) \phi^{p16}(X_3) R/K^{p20}$ motif. (b) Peptide ligand sequences of MRGPRX1. Sequence comparisons of several peptide-based allergens with similarities to the $\phi^{B17}(X_{1-2}) R^{B20} \phi^{B21}$ motif. (c) Effects of BAM8-22, CNF-Tx2, γ 1-MSH, hemoglobin β -chain (34-42), P60 (part of C5orf29) induced MRGPRX1 activation evaluated via Gai-Gy dissociation assay. Data from three independent experiments are presented as the mean \pm SEM (n=3). All data were analyzed by two-sided one-way ANOVA with Turkey test. (d) Effects of different BAM8-22 mutations on BAM8-22 induced Gai-Gy dissociation. Data from three independent experiments are presented as the mean \pm SEM (n=3). All data were analyzed by two-sided one-way ANOVA with Turkey test. (e) Effects of different CNF-Tx2 mutations on CNF-Tx2 induced Gai-Gy dissociation. Data from three independent experiments are presented as the mean \pm SEM (n=3). All data were analyzed by two-sided one-

way ANOVA with Turkey test.

2) The authors use a MRGPRX1 fused at its N terminus to BRIL for structural studies. Could the authors comment on whether this construct behaves similarly to BRIL-free MRGPRX1 in pharmacological assays? This comparison is important for understanding structural data versus pharmacological data.

Reply: We thank the reviewer for his/her helpful suggestions. To increase receptor expression, thermostabilized cytochrome b562RIL (BRIL) was incorporated at the N-terminus of full-length MRGPRX1. The fusion of the BRIL to the N-terminal of MRGPRX1 showed no significant effects on G protein coupling activity of MRGPRX1 (Fig. R2 and Supplementary Fig. 1 in the revised manuscript).

Fig. R2 (also shown in Supplementary Fig.1) Purification of MRGPRX1-Gi1-scFv complex.

(a-b) Dose-response curves of the BAM8-22 induced $G_{\alpha i}$ - G_{γ} dissociation (a) and $G_{\alpha q}$ - G_{γ} dissociation (b) in MRGPRX1-WT or BriI-MRGPRX1 overexpressing cells. Data from three independent experiments are presented as the mean \pm SEM (n=3). (c) Dose-response curves of the CNF-Tx2 induced $G_{\alpha i}$ - G_{γ} dissociation in MRGPRX1-WT or BriI-MRGPRX1 overexpressing cells. Data from three independent experiments are presented as the mean \pm SEM (n=3). (d) Left panel: representative elution profile of BAM8-22-MRGPRX1-Gi1-scFv16

complex. BAM8-22-MRGPRX1-Gi1-scFv16 complex on Superose 6 Increase 10/300 column and SDS-PAGE of the size-exclusion chromatography peak. Right panel, coomassie-stained PAGE of the isolated peak fraction from the Superose 6. (e) Left panel: representative elution profile of BAM8-22-MRGPRX1-Gq-scFv16 complex. BAM8-22-MRGPRX1-Gq-scFv16 complex on Superose 6 Increase 10/300 column and SDS-PAGE of the size-exclusion chromatography peak. Right panel, coomassie blue-stained PAGE of the isolated peak fraction from the Superose 6. (f) Left panel: representative elution profile of CNF-Tx2-MRGPRX1-Gi1-scFv16 complex. CNF-Tx2-MRGPRX1-Gi1-scFv16 complex on Superose 6 Increase 10/300 column and SDS-PAGE of the size-exclusion chromatography peak. Right panel, coomassie-blue stained PAGE of the isolated peak fraction from the Superose 6.

3) The signaling properties of many ligands and their mutants (as well as receptor mutants) is described with a single pharmacological parameter (pEC_{50}). This seems to leave out important information about maximal responses (efficacy). Addition of this information would be helpful for evaluating the impact of mutations on signaling.

Reply: We thank the reviewer for his/her helpful suggestions. We have added the E_{max} data according to this reviewer's suggestion in the revised manuscript in Fig. 2c-2d, Supplementary Fig. 9a, Supplementary Fig. 9d, Supplementary Fig. 11c and Supplementary Fig. 11g as follows:

Fig. 2 The BAM8-22 binding pocket in MRGPRX1-Gi1/Gq complexes.

(a) The “U”-shaped binding pose of BAM8-22 in MRGPRX1-Gi1/Gq complexes. (b) Three-dimensional (3D) representation of the detailed interactions between BAM8-22 and MRGPRX1 in MRGPRX1-Gq complex. (c) Effects of different mutations on BAM8-22 of MRGPRX1 induced G α i-G γ dissociation. The maximal response (Emax) and EC50 derived from the dose-response curve is shown. Statistical differences between MRGPRX1 WT and mutations were presented as the mean \pm SEM of three independent experiments and determined by two-sided one-way ANOVA with Tukey test. *, P<0.05; **, P<0.01; ***, P<0.001; n.s., no significant difference; ND, not detected. (P=0.0218, 0.0062, 0.0005, 0.0001, ND, 0.0004,

0.0002, 0.0001, <0.0001, 0.5044, 0.0548 from top to bottom of Δ pEC50). **(d)** Effects of different mutations within the ligand-binding pocket of MRGPRX1 on BAM8-22 induced Gai-Gy dissociation. The maximal response (Emax) and EC50 derived from the dose-response curve is shown. Statistical differences between MRGPRX1 WT and mutations were presented as the mean \pm SEM of three independent experiments and determined by two-sided one-way ANOVA with Tukey test. *, P<0.05; **, P<0.01; ***, P<0.001; n.s., no significant difference; ND, not detected. (P=0.0218, 0.0062, 0.0005, 0.0001, ND, 0.0004, 0.0002, 0.0001, <0.0001, 0.5044, 0.0548 from top to bottom of Δ pEC50). **(e)** Heatmap of pairing of BAM8-22 mutants with MRGPRX1 WT and MRGPRX1 alanine scanning mutants. The receptor mutants that did not show significantly decreased EC50 values compared to those of the WT receptor when binding to a specific BAM8-22 mutant are highlighted by red color. **(f)** Structural representation of M15 and Y17 in MRGPRX1-Gai/Gq complexes respectively. The distance was depicted as the dashed red line. **(g)** Effects of F236^{6.55}, R246^{ECL3} and L249^{7.30} on BAM8-22 induced Gai/Gq activity by Gai-Gy dissociation assay. The curve data from three independent measurements are measured as mean \pm SEM (n=3). All data were determined by two-sided one-way ANOVA with Tukey test. ***, P < 0.001, n.s., no significant difference.

Supplementary Fig. 9 Binding of CNF-Tx2 to MRGPRX1.

(a) Emax effects of different mutations on CNF-Tx2 of MRGPRX1 induced G α i-G γ dissociation. The maximal response (Emax) is presented as the mean \pm SEM of three independent experiments. Statistical differences between MRGPRX1 WT and mutations were determined by two-sided one-way ANOVA with Tukey test. *, P<0.05; **, P<0.01; ***, P<0.001; n.s., no significant difference; ND, not detected. (P= P<0.001, ND, 0.6262, 0.109, P<0.001 from left to right). **(b)** Effects of different CNF-Tx2 mutations on CNF-Tx2 induced G α i-G γ dissociation in MRGPRX1 overexpressing HEK293 cells. The curve data from three independent measurements are measured as mean \pm SEM (n=3). **(c)** Three-dimensional (3D) representation of the detailed interactions between CNF-Tx2 and MRGPRX1 in MRGPRX1-Gi1-model2 complex. **(d)** Emax effects of different mutations within the ligand-binding pocket of MRGPRX1 on CNF-Tx2 induced G α i-G γ dissociation. Statistical differences between MRGPRX1 WT and mutations were presented as the mean \pm SEM of three independent experiments and determined by two-sided one-way ANOVA with Tukey test. *, P<0.05; **, P<0.01; ***, P<0.001; n.s., no significant difference; ND, not detected. (P=0.3501, 0.1541, ND, ND, <0.001, 0.0041, ND, <0.001, 0.2487, <0.001 from left to right). **(e-f)** Effects of different mutations within the ligand-binding pocket of MRGPRX1 on CNF-Tx2 induced G α i-G γ dissociation in MRGPRX1 overexpressing cells. The curve data from three independent measurements are measured as mean \pm SEM (n=3).

Supplementary Fig.11 Coupling of MRPGRX1 with Gi and Gq.

(a) Detailed interactions between the TM bundles of MRGPRX1 and the $\alpha 5$ -helix end of G α i. (b) Detailed interactions between the ICL2 of MRGPRX1 and the G α i. (c) Emax effects of different mutations of G protein interface mutations of MRGPRX1 on G α i. Statistical differences between MRGPRX1 WT and mutations were presented as the mean \pm SEM of three independent experiments and determined by two-sided one-way ANOVA with Tukey test. *, P<0.05; **, P<0.01; ***, P<0.001; n.s., no significant difference; ND, not detected. (P=<0.001, <0.001, ND, ND, ND, <0.001, <0.001, ND from top to bottom). (d) The effects of G protein interface mutations of MRGPRX1 on G α i. The curve data from three independent measurements are measured as mean \pm SEM (n=3). (e) Detailed interactions between the bulky end of $\alpha 5$ helix of G α q and the V124^{3,54}, L198^{5,57}, R213^{6,32}, L214^{6,33} and T217^{6,36} of MRGPRX1. (f) Detailed interactions between the E357^{G.H5.22}, Y358^{G.H5.23} of G α q and Y64^{2,42}, F61^{2,39} and Y130^{ICL2}, R131^{ICL2} of MRGPRX1. (g) Emax effects of different mutations of G protein interface mutations of MRGPRX1 on G α q. Statistical differences between MRGPRX1 WT and mutations were presented as the mean \pm SEM of three independent experiments and determined by two-sided one-way ANOVA with Tukey test. *, P<0.05; **, P<0.01; ***, P<0.001; n.s., no significant difference; ND, not detected. (P=0.0007, <0.001, 0.0222, <0.001, <0.001, <0.001, 0.0005, <0.001, ND, ND, <0.001 from top to bottom). (h) The effects of G protein interface mutations of MRGPRX1 on G α q. The curve data from three independent measurements are measured as mean \pm SEM (n=3).

4) Receptor signaling through Gi and Gq transducers is evaluated. It seems that some ligands show differential signaling through these pathways. There is a great deal of interest in ligands that show functional selectivity for signaling through different pathways. It would be helpful to see these differences quantified through ligand bias calculations.

Reply: We thank the reviewer for his/her helpful suggestions. To facilitate better visualization of the bias, we have included representative concentration-response data and made side-by-side comparisons of the ligands of particular signaling (effector) pairs (Fig. R3a-R3b). Then we calculated the ligand bias factor according to the operational model that described previously by Sudarshan Rajagopal and Pro. Lefkowitz et al. (*Mol Pharmacol.* 2011 80(3):367-77.) (Fig.

R3c). The β value was calculated according to formulas.

$$\beta = \log \left(\left[\frac{E_{\max,P1}}{EC_{50,P1}} \frac{EC_{50,P2}}{E_{\max,P2}} \right]_{\text{lig}} \times \left[\frac{E_{\max,P2}}{EC_{50,P2}} \frac{EC_{50,P1}}{E_{\max,P1}} \right]_{\text{ref}} \right)$$

P1: $G\alpha_i$ -G γ dissociation assay data, P2: $G\alpha_q$ -G γ dissociation assay data, $\beta > 0$, Gi biased; $\beta < 0$, Gq biased.

Therefore, as shown in the Fig. R2, chloroquine (CQ) is a Gq bias ligand, whereas CNF-Tx2 showed Gi bias when we compared Gq activation over Gi using BAM8-22 as a reference.

Fig. R3 Ligands induced different G protein activation downstream of MRGPRX1.

(a) Concentration-dependent response curves of MRGPRX1 in response to ligands by $G\alpha_i$ -G γ dissociation assay. Values are mean \pm SEM from three independent experiments (n=3) performed in triplicates. (b) Concentration-dependent response curves of MRGPRX1 in response to ligands by $G\alpha_q$ -G γ dissociation assay. Values are mean \pm SEM from three independent experiments (n=3) performed in triplicates. (c) Comparison of the biased properties of CNF-Tx2 and Chloroquine. Both CNF-Tx2 and Chloroquine were assessed for Gi signaling (a) and Gq signaling (b). The bias factor (β value) of CNF-Tx2 and Chloroquine was calculated using BAM8-22 as the reference.

5) For comparing the two different models of CNF-Tx2 interaction with receptor (Figure 3A) it seems that deleting the C terminal VRI motif would allow assessment of the importance of these residues for receptor activation. Is there a reason not to test this directly?

Reply: We thank the reviewer for his/her helpful suggestions. As suggested by the reviewer, we have synthesized the truncated CNF-Tx2 peptide (deleting the C terminal VRI motif) and

found that the truncation almost abolished the activity of MRGPRX1 compared to the CNF-Tx2 wide-type peptide (Fig. R4, also shown in Fig. 3d and Supplementary Fig. 9a-9b in the revised manuscript).

Fig. R4 (also shown in Fig. 3d and Supplementary Fig. S9a-S9b).

Effects of the wide-type and truncated CNF-Tx2 peptides induced G α i-G γ dissociation in MRGPRX1 overexpressing HEK293 cells. The curve data from three independent measurements are measured as mean \pm SEM (n=3). ND, not detectable due to low signal.

Minor comments:

6) Many of the ligand residues do not make contact with the receptor directly. Based on the structure activity relationship studies in the manuscript can the authors comment a bit more on the role of ligand residues not directly involved in receptor binding?

Reply: We thank the reviewer for his/her helpful suggestions. Based on our solved structure, the interactions of some residues are not observed, because the sidechains are not modeled due to ambiguous poor EM density, such as E12, D16, K19 of BAM8-22 in BAM8-22-MRGPRX1-Gi complex and R10, P11, E12, K19 of BAM8-22 in BAM8-22-MRGPRX1-Gq complex. These residues may directly interact with MRGPRX1 but we didn't observe these interactions.

In addition, the binding energy is the sum of entropy and enthalpy. Whereas interactions contributed to the enthalpy, the entropy is mostly dependent on conformations of ligands and receptors. We speculated that several residues of BAM8-22 played important roles in entropy changes when the BAM8-22 binds to MRGPRX1, therefore contributing to the binding energy of the peptide without direct interactions with the MRGPRX1. We have incorporated these discussions in the “Binding of BAM8-22 to MRGPRX1” of the revised manuscript.

7) The interaction of peptidic ligands with receptors such as GPCRs is an active area of inquiry for newly developed modeling algorithms. Could the authors perform a simple comparison of experimental structures with those produced by a modeling algorithm such as Alphafold2? Either success or failure of Alphafold2 in predicting structural details would be informative.

Reply: We thank the reviewer for his/her helpful suggestions. As far as we know, Alphafold2 can currently only be used to predict protein structure but not protein-protein interactions. A recent article that has not yet been officially published mentioned that AlphaFold version 2.3 was explicitly trained to model protein-protein interactions, but this program requires large amounts of physical space, and we are still working on it. In addition, using the ZDOCK program, which is commonly used to predict the binding patterns of peptide ligands to receptors, we found that the top two predicted results were quite different from the binding patterns of the ligands in our resolved structures (Fig. R5), so we speculated the predictions at the current stage were still not comparable to experimental data.

Fig. R5 The structural representation of ligand binding models of two highest-scoring clusters simulated by ZDOCK in CNF-Tx2-MRGPRX1-Gi complex.

8) On line 52: “GPCRs” should be “GPCR”

Reply: We thank the reviewer for his/her helpful suggestions, we have replaced the “GPCRs” with “GPCR” in our revised manuscript:

Line 45 - ‘MRGPRX1, a Mas-related GPCR (MRGPR), is a key receptor for itch perception, and targeting MRGPRX1 may have the potential to treat both chronic itch and pain.’

9) On line 91 the language used is confusing. Saying that GPCRs are known to couple to TRPA1 is not clear. GPCRs typically couple to G proteins and beta-arrestin. Do these GPCRs couple to TRPA1 in an analogous way or is TRPA1 a downstream response?

Reply: We thank the reviewer for his/her helpful suggestions. We have replaced the term of ‘couple to’ with ‘**functionally link to**’ and ‘**link to**’ according to this reviewer’s suggestion in the revised manuscript, which includes the following places:

Line 82 - ‘The functional homologs of MRGPRX1 in mice, MrgprA3 and MrgprC11, are known to **functionally link to** TRPA1, which is essential for itch sensation.’

Line 83- ‘These two receptors are **functionally link to** TRPA1 through different mechanisms. Whereas MrgprC11 connected to TRPA1 through Gq-PLC signaling, MrgprA3 was found to **link to** TRPA1 through G β γ . Using dental afferents of human samples, MRGPRX1 was shown to sensitize TRPA1 and instigate membrane depolarization^{10, 11}.’

10) On line 224: “was” should be deleted

Reply: We thank the reviewer for his/her helpful suggestions, we have deleted the word ‘was’ on **line 214** in our revised manuscript.

Reviewer #2 (Remarks to the Author):

The manuscript by Guo et al., reports the cryo-EM structures of MRGPRX1-Gi1 in complex

with BAM8-22 or CNF-Tx2 and MRGPRX1-Gq in complex with BAM8-22, revealing a unique shallow ligand binding pocket at the extracellular ends of TM3 and TM4 for peptidic allergen recognition. They also describe the conserved kink motif present in the MRGPRX family for MRGPRX1 activation. In addition, they reveal both the Gi1 and Gq coupling mechanisms of MRGPRX1 and found that TM3 and ICL2 of MRGPRX1 form specific interactions with the bulky end of the $\alpha 5$ helix of G α q contributed to most of the specific G α q coupling mechanisms. These observations are nicely verified by their mutagenesis studies. Overall, the manuscript describes elegant and rigorous structural analysis and biochemical experiments. Their maps look like they are good quality. The mechanism that is proposed is reasonable and is based on well-designed experiments that are suggested by the structure. However, before publication could be recommended, a few, mostly minor, issues should be addressed:

Reply: We thank the reviewer for his/her positive comments.

1) As far as I know, the structure of Gq-coupled MRGPRX1 with BAM8-22 has been already reported (PDB 8DWC). The authors should compare with their structure and state the differences and similarities.

Reply: We thank the reviewer for his/her helpful suggestions. Following the reviewer's suggestions, we have compared our structure with the recently reported structure of Gq-coupled MRGPRX1 bound with BAM8-22 (PDB ID: 8DWC reported by Yongfeng Liu et al.) and added a figure (Fig. R6 and Supplementary Fig. 13 in the revised manuscript) in our revised manuscript. Overall, BAM8-22 showed similar conformations in these two structures with an average RMSD of 0.9 Å (Fig. R6a and Supplementary Fig. 13a in the revised manuscript). However, compared with Gq in 8DWC structure, the orientation of $\alpha 5$ Helix of Gq in our structure shifted approximately by 4.1Å (Fig. R6b and Supplementary Fig. 13b in the revised manuscript). This difference may be due to the different Gq construct used in the complex formation (they used a mini-G α q β N chimera. vs. we used a modified G α q β N chimera). For the peptide ligand, our EM density for BAM-22 was continuous when we set up the contour level

at 0.13V. In contrast, the EM density for BAM-22 in their structure (PDB ID: 8DWC) showed broken places when we set up the contour level at 0.043V. In detail, the cryo-EM density of endogenous BAM8-22 in Gq-coupled MRGPRX1 complex (PDB ID: 8DWC) was broken at the locations of R20 and Y21 residue, whereas in our structure these positions had continuous EM density which enabled unambiguous assignment of these two residues (Fig. R6c and Supplementary Fig. 13c in the revised manuscript).

Fig. R6 (also shown in Supplementary Fig. 13) Comparison of the two structures of BAM8-22-MRGPRX1-Gq complexes (we solved vs. PDB ID: 8DWC).

(a) The root-mean-square-deviation (RMSD) of BAM8-22 between BAM8-22-MRGPRX1-Gq and 8DWC complex structures. (b) Comparison of the G protein between the newly solved BAM8-22-MRGPRX1-Gq structure (green) and the structure solved by Yongfeng Liu et al. (PDB ID: 8DWC). (c) Comparison of ligand densities in the BAM8-22-MRGPRX1-Gq and 8DWC structures.

2) Considering the structure of MRGPRX4-Gq complex has been resolved, it would be better if authors could compare their differences and similarities.

Reply: We thank the reviewer for his/her helpful suggestions. Here, by comparing our structure with the structure of MS47134-MRGPRX4-Gq complex resolved by Bryan L. Roth et al. (PDB ID: 8DWC), we observed a marked difference in the upper half of the TM bundles and both TM4 and TM7 undergo clockwise rotations when we aligned the TM6 together (Fig. R7a and Supplementary Fig. 14a in the revised manuscript). Moreover, the locations of the ligands in MRGPRX1 and MRGPRX4 are different (Fig. R7b and Supplementary Fig. 14b in the revised manuscript). Notably, the MS47134 occupied a deeper place in MRGPRX4 compared with the position of BAM8-22 in the BAM8-22-MRGPRX1-Gq complex. In the interface between the receptors and Gq, the $\alpha 5$ helical end of G α q in the MRGPRX4-Gq complex exhibited an approximately 2 Å downward shift compared with that of MRGPRX1-Gq complex (Fig. R7c and Supplementary Fig. 14c in the revised manuscript). Gq interacts with 19 residues of MRGPRX4, whereas interacts with 23 residues of MRGPRX1 (Fig. R7d and Supplementary Fig. 14d in the revised manuscript).

Fig. R7 (also shown in Supplementary Fig. 14) Structural comparison of MRGPRX1 with MRGPRX4.

(a) A cytoplasmic view of the BAM8-22-MRGPRX1 7TM bundle compared with MS47134-MRGPRX4 (PDB ID: 7S8P). MRGPRX1 is shown in salmon, MRGPRX4 in light sky blue. (b) Three-dimensional (3D) representation of BAM8-22 in the MRGPRX1 and MS47134-MRGPRX4 (PDB ID: 7S8P). BAM8-22 is shown in cyan, MS47134 in hot pink. (c) The structural representation and comparison of the interfaces between the MRGPRX1-Gq and MRGPRX4-Gq complexes. Ribbon representation: Gq bound to MRGPRX1 is shown in yellow, Gq bound to MRGPRX4 is shown in gray. (d) Comparison of the Gq coupling interfaces in cryo-EM structures of BAM8-22-MRGPRX1-Gq, and MS47134-MRGPRX4 (PDB ID: 7S8P) complexes. Residues of MRGPRX1 in contact with Gq were illustrated as green dots.

3) Page 3, line 52: “MGRPRX1” “MRGPRX1”

Reply: We thank the reviewer for his/her helpful suggestions. We have revised it in our revised manuscript as follows:

Line 45: ‘MRGPRX1, a Mas-related GPCR (MRGPR), is a key receptor for itch perception, and targeting MRGPRX1 may have the potential to treat both chronic itch and pain.’

4) Page 4, line 91 and 93 “Mrgprc11” “MrgprC11”

Reply: We thank the reviewer for his/her helpful suggestions. We have revised it in our revised manuscript as follows:

Line 81: ‘The functional homologs of MRGPRX1 in mice, MrgprA3 and MrgprC11, are known to couple to TRPA1, which is essential for itch sensation. These two receptors are coupled to TRPA1 through different mechanisms.’

5) Page 5, line 107, 121 and 123: “2.9Å” “2.98Å” “2.8Å” “2.84Å”

Reply: We thank the reviewer for his/her very helpful suggestions. We have uniformly reserved one decimal place for the resolution in our revised manuscript and figure as follows:

Line 92: ‘To understand the structural basis of the itch sensation, particularly in the sensation processes of the peripheral nervous system, and to develop important therapeutic tools for the treatment of itch-related diseases and pains in the central nervous system, we determined the structures of MRGPRX1-Gi1 in complex with bovine adrenal medulla 8-22 (BAM8-22) or CNF-Tx2 and the structure of MRGPRX1-Gq in complex with BAM8-22 at the resolutions of 3.0 Å, 2.8Å, 2.9Å, respectively.’

6) Page 9, line 218: “is fits” “fits”

Reply: We thank the reviewer for his/her helpful suggestions. We have revised it in our revised manuscript as follows:

Line 206: “Compared with mode 2 the CNF-Tx2 in mode 1 fits better with EM density. We then performed a molecular dynamics simulation by including side chain atoms that were not defined by EM density and the result indicated that model 1 was more stable (Fig. 3e).”

7) Page 9, line 224: There is no figure provided to fit the description of the interactions in mode 2.

Reply: We thank the reviewer for his/her helpful suggestions. We have added Fig. R8 (also shown in Supplementary Fig. 9c) to fit the description of the interactions in mode 2: “Compared with mode 1, the CNF-Tx2 in mode 2 was lost specific interactions with E157^{4,60} and D177^{5,36} and formed new contact with F239^{6,58}.”

Fig. R8 (also shown in Supplementary Fig. 9c) Three-dimensional (3D) representation of the detailed interactions between CNF-Tx2 and MRGPRX1 in MRGPRX1-Gi1-mode2 complex.

8) Please check “Cryo-EM data acquisition” again: The pixel size does not fit the pixel size given in Supplementary Table 1, as well as defocus range and total exposure electron. Meanwhile a dose rate of about 7.8 electrons per Å² per second and total exposure time of 8 s should be the parameters of K2 camera other than K3, the authors need clarify the detailed data collection parameters for each structure.

Reply: We thank the reviewer for his/her helpful suggestions. We have carefully re-examined the structural information and have revised the Method section and Supplementary Table 1 to ensure accuracy and consistency:

Cryo-EM data acquisition

The purified BAM8-22-MRGPRX1-Gi1 complex (3.0 µl) at 5.0 mg/ml, BAM8-22-MRGPRX1-Gq complex (3.0 µl) at 4.0 mg/ml and the CNF-Tx2-MRGPRX1-Gi1 complex (3.0 µl) at 4.5 mg/ml were applied onto a glow-discharged holey carbon grid (Quantifoil R1.2/1.3), and subsequently vitrified using a FEI Vitrobot Mark IV (Thermo Fisher Scientific). The cryo-grids were initially screened at a nominal magnification of ×92,000 in an FEI Talos Arctica microscope (200 kV), equipped with an FEI Ceta camera. High-quality grids were transferred to a FEI Titan Krios electron microscope (Thermo Fisher Scientific) equipped with a Gatan K2 or K3 Summit direct electron detector and a Gatan Quantum-LS Energy Filter (GIF, slit width of 20 eV).

For the BAM8-22-MRGPRX1-Gq complex dataset, 5,601 movies were collected on a Titan Krios equipped with a Gatan K3 direct electron detection device at 300 kV with a magnification of 81,000, corresponding to a pixel size 1.04 Å. We collected a total of 36 frames accumulating to a total dose of 50 e⁻/Å² over 2.5 s exposure on each TIF format movie.

For the CNF-Tx2-MRGPRX1-Gi1 complexes, 3,085 movies were collected on a Titan Krios equipped with a Gatan K2 direct electron detection device at 300 kV with a magnification of 130,000, corresponding to a pixel size 1.08 Å. The total exposure time was 8 s, resulting in

an accumulated dose of 50 electrons per Å² and a total of 32 frames per movie.

For the BAM8-22-MRGPRX1-Gi1 complexes, all 5,540 movies were collected on a Titan Krios equipped with a Gatan K3 direct electron detection device at 300 kV with a magnification of 130,000, corresponding to a pixel size 0.89 Å. The total exposure time was 3 s, resulting in an accumulated dose of 60 electrons per Å² and a total of 32 frames per movie.

Supplementary Table 1: Cryo-EM Data Collection, Model Refinement, and Validation

Statistics

Complex	BAM8-22- MRGPRX1-Gq	CNF-Tx2- MRGPRX1-Gi	BAM8-22- MRGPRX1-Gi
Data Collection and Processing			
Magnification	81000	130000	130000
Voltage (kV)	300	300	300
Electron exposure (e ⁻ /Å ²)	50	50	60
Defocus range (µm)	-1.2 to -2.2	-0.8 to -1.2	-0.8 to -1.2
Pixel size (Å)	1.04	1.08	0.89
Symmetry imposed	C1	C1	C1
Initial particle projections (no.)	11,127,531	146,824,9	3,628,139
Final particle projections (no.)	1,316,443	315,448	925,644
Map resolution (Å)	2.9	2.8	3.0
FSC threshold	0.143	0.143	0.143
Map resolution range (Å)	2.5-5	1.9-6.5	2.5-7
Refinement			
Initial model used (PDB accession number)	7UVY	7UVY	7UVY
Model resolution (Å)	3.1	3.0	3.2
FSC threshold	0.5	0.5	0.5
Model Composition			

Peptide chains	6	6	6
Protein residues	1101	1062	1076
Ligand	1	1	1
B factors (Å ²)			
Protein	27.93	50.79	92.83
RMSD			
Bond lengths (Å)	0.008	0.010	0.007
Bond angles (°)	1.105	1.384	1.145
Validation			
MolProbity score	1.78	2.16	1.92
Clashscore	8	7	9
Rotamer outliers (%)	0.1	0	0.7
Ramachandran Plot			
Favored (%)	95.74	93.82	94.30
Allowed (%)	4.26	6.18	5.70
Disallowed (%)	0	0	0
PDB accession number	8JGF	8JGB	8JGG
EMDB accession number	EMD-36232	EMD-36229	EMD-36233

9) The resolution showed in Supplementary Fig. S2c is 2.85Å which is not consistent with the labeled 2.7Å.

Reply: We thank the reviewer for his/her helpful suggestions. We have checked the single particle reconstruction of the BAM8-22-MRGPRX1-Gq complex, and revised the resolution to 2.9 Å in Supplementary Fig. 2c:

Supplementary Figure 2. Cryo-EM images and single particle reconstruction of the BAM8-22-MRGPRX1-Gq complex.

(a) Flow chart for cryo-EM data processing of BAM8-22-MRGPRX1-Gq complex. (b) Representative Cryo-EM micrograph of BAM8-22-MRGPRX1-Gq complex (left) and 2D class averages (right). (c) Fourier shell correlation curves for the final 3D density maps of BAM8-22-bound MRGPRX1-Gq complex. At the Fourier shell correlation (FSC) 0.143 cut-off, the overall resolution was 2.9 Å. (d) 3D density map colored according to local resolution (Å) of the BAM8-22-MRGPRX1-Gq trimer complex.

10) In Supplementary Fig. S2d, local resolution of the GPCR part seems unreasonable.

Reply: We thank the reviewer for his/her very helpful suggestions. We have replaced the new 3D density map colored according to the local resolution (Å) of the BAM8-22-MRGPRX1-Gq trimer complex as shown above.

11) Fig. 2c, Supplementary Fig. S2a, b : R20K R20A

Reply: We thank the reviewer for his/her helpful suggestions. Due to the poor solubility of peptide BAM8-22-R20A, we used BAM8-22-R20K for the activity pairing assays of BAM8-12 mutants. We have included the reason why we used R20K replacing 20A in the Figure2 legend of the revised manuscript.

12) The clashscore of three structures in Supplementary Table 1 is not consistent with the validation reports.

Reply: We thank the reviewer for his/her helpful suggestions. We have refined all the models and revised these issues in the revised model, and the clashscore of three structures in Supplementary Table 1 has been consistent with the formal version of validation reports.

13) According to the information of 5.3.2 protein sidechains in validation reports, more efforts need to be done to correct outliers. And it would be better to provide formal version of validation reports next time.

Reply: We thank the reviewer for his/her helpful suggestions. We have examined and revised the side chain outliers in BAM8-22-MRGPRX1-Gi, BAM8-22-MRGPRX1-Gq and CNF-Tx2-MRGPRX1-Gi structures, respectively. A formal version of validation reports has been included in the revised submission.

Reviewers' Comments:

Reviewer #1:

Remarks to the Author:

The authors have provided new data and discussion to effectively address my previous comments. There are a few small follow up questions for which I request further responses.

- For the new Emax data (Comment 3) it was striking that almost all mutations cause a reduction in Emax relative to the wild type receptor. It can be tricky to interpret Emax data as differences in either receptor functionality or receptor localization can lead to variation in Emax. Could the authors provide a discussion on why there seem to be such dramatic effects on Emax levels? Are mutations known to affect receptor stability or trafficking?
- The new analysis of ligand bias (comment 4) is interesting but it seems only to be included as a figure for review. I think this should be included as a supporting figure unless there is a compelling reason to exclude it.
- There has been extensive analysis of protein-protein interactions (comment 7) applying modeling from AlphaFold2 in AlphaFold multimer (PMID: 35900023). Web tools such as Colabfold (<https://colab.research.google.com/github/sokrypton/ColabFold/blob/main/AlphaFold2.ipynb>) make this straightforward. This approach has been applied to other GPCR/ligand pairs (<https://pubmed.ncbi.nlm.nih.gov/37092865/>). I think a comparison of experimental data with AlphaFold2-based modeling results would be of wide interest.

Reviewer #2:

Remarks to the Author:

The authors have made an effort to address my concerns. However I still have two minor questions that should be addressed before supporting publication of the revised manuscript.

1. The local resolution map showed in Supplementary Fig 2d still seems incorrect and is not consistent with the Supplementary table 1 (map resolution range). The authors should check it carefully.
2. The resolution of BAM8-22- MRGPRX1-Gq provided in validation report is 2.7Å which is different from what was mentioned in the manuscript. And the validation reports of these three structures here are still informal version.

REVIEWERS' COMMENTS

Reviewer #1 (Remarks to the Author):

The authors have provided new data and discussion to effectively address my previous comments. There are a few small follow up questions for which I request further responses.

Reply: We thank the reviewer for his/her positive comments.

1. For the new Emax data (Comment 3) it was striking that almost all mutations cause a reduction in Emax relative to the wild type receptor. It can be tricky to interpret Emax data as differences in either receptor functionality or receptor localization can lead to variation in Emax. Could the authors provide a discussion on why there seem to be such dramatic effects on Emax levels? Are mutations known to affect receptor stability or trafficking?

Reply: We thank the reviewer for his/her helpful suggestions. We have re-examined the expression level of wild type and mutants on the cell membrane and the Emax values were updated after revision of the transfecting plasmid amounts to enable similar plasma membrane expression of the wild type or mutant MRGPRX1 receptors. As results, several mutants were not responsive in response to BAM-22 or CNF stimulation. Approximately 20%~30% mutants showed significant effects on Emax, whereas more than 60% of mutants showed no significant change for Emax. The EC50 didn't show significant changes compared with previous version. We have included the new data in the revised manuscript in Fig. 2d, Supplementary Fig. 8c, Supplementary Fig. 10d-10f, Supplementary Fig. 12c-12d and Supplementary Fig. 12g-12h as follows:

Fig. R1 (also shown in Fig. 2) The BAM8-22 binding pocket in MRGPRX1-Gi/Gq complexes. **a**, The “U”-shaped binding pose of BAM8-22 in MRGPRX1-Gi/Gq complexes. **b**, Three-dimensional (3D) representation of the detailed interactions between BAM8-22 and MRGPRX1 in MRGPRX1-Gq complex. **c**, Effects of different BAM8-22 mutations on BAM8-22 induced G_{ai} - G_{γ} dissociation. Bar graph for EC_{50} was presented. Due to the poor solubility of peptide BAM8-22-R20A, we used BAM8-22-R20K for the activity pairing assays of BAM8-22 mutants. Statistical differences between BAM8-22 WT and mutations were determined by two-sided one-way ANOVA with Tukey test. *, $P < 0.05$; **, $P < 0.01$; ***, $P < 0.001$; n.s., no

significant difference. (P=<0.001, <0.001, <0.001, <0.001, <0.001, <0.001, <0.001, <0.001, 0.9999, <0.001, 0.7568, 0.6225 from top to bottom of ΔpEC_{50} ; P=<0.001, <0.001, <0.001, <0.001, 0.0003, 0.0002, <0.001, 0.0002, 0.2383, <0.001, 0.9959, 0.5178 from top to bottom of E_{max}). Data from three independent experiments are presented as the mean \pm SEM (n=3). **d**, Effects of different mutations within the ligand-binding pocket of MRGPRX1 on BAM8-22 induced Gai-G γ dissociation. Statistical differences between MRGPRX1 WT and mutations were determined by two-sided one-way ANOVA with Tukey test. *, P<0.05; **, P<0.01; ***, P<0.001; ns, no significant difference; ND, not detected. (P=0.7609, 0.0117, <0.001, <0.001, ND, ND, <0.001, 0.0003, <0.001, <0.001, 0.9616 from top to bottom of ΔpEC_{50} ; P=0.9916, 0.9642, 0.1755, 0.2536, ND, ND, <0.001, 0.4812, 0.2536, <0.001, 0.1643 from top to bottom of E_{max}). Data from three independent experiments are presented as the mean \pm SEM (n=3). **e**, Heatmap of pairing of BAM8-22 mutants with MRGPRX1 WT and MRGPRX1 alanine scanning mutants. The receptor mutants that did not show significantly decreased EC50 values compared to those of the WT receptor when binding to a specific BAM8-22 mutant are highlighted by red color. **f**, Structural representation of M15 and Y17 in MRGPRX1-Gai/Gq complexes respectively. The distance was depicted as the dashed red line. **g**, Effects of F236^{6,55}, R246^{ECL3} and L249^{7,30}-on BAM8-22 induced Gai/Gq activity by Gai-G γ dissociation assay. The curve data from three independent measurements are measured as mean \pm SEM (n=3). All data were determined by two-sided one-way ANOVA with Tukey test. ***, P < 0.001, n.s., no significant difference.

Fig. R2 (also shown in Supplementary Fig. 8) Effects of different mutations in BAM8-22 or different mutations within the ligand-binding pocket of MRGPRX1 induced Gai-Gy and Gαq-Gy dissociation.

(a) Effects of different BAM8-22 mutations on BAM8-22 induced Gai-Gy dissociation in MRGPRX1 overexpressing HEK293 cells. The curve data from three independent measurements are measured as mean \pm SEM (n=3). (b) Effects of different BAM8-22 mutations on BAM8-22 induced Gαq-Gy dissociation in MRGPRX1 overexpressing HEK293 cells. The curve data from three independent measurements are measured as mean \pm SEM (n=3). (c) Effects of different mutations within the ligand-binding pocket of MRGPRX1 on BAM8-22

induced G α i-G γ dissociation in MRGPRX1 overexpressing cells. The curve data from at least three independent measurements are measured as mean \pm SEM (n=3). **(d)** Effects of different mutations within the ligand-binding pocket of MRGPRX1 on BAM8-22 induced G α i-G γ dissociation and G α q-G γ dissociation in MRGPRX1 overexpressing cells. The curve data from three independent measurements are measured as mean \pm SEM (n=3).

Fig. R3 (also shown in Supplementary Fig. 10) Binding of CNF-Tx2 and BAM8-22 to MRGPRX1.

(a) Comparison of CNF-Tx2 binding modes simulated by Colabfold and in CNF-Tx2-MRGPRX1-Gi1 complex structure that our resolved. The CNF-Tx2 model 1-5 predicted by Colabfold are shown in tangold, skybluemedium slate blue, plummedium turquoise, cornflower

bluelightgreen and dark salmonwheat, CNF-Tx2 in our resolved CNF-Tx2-MRGPRX1-Gi1 complex is shown in red. **(b)** Comparison of BAM8-22 binding modes simulated by Colabfold and in BAM8-22-MRGPRX1-Gi1 complex structure that our resolved. The BAM8-22 model 1-5 predicted by Colabfold are shown in plum, dark green, light sky blue, medium purple and dark orange, BAM8-22 in our resolved BAM8-22-MRGPRX1-Gi1 complex is shown in red. **(c)** Comparison of BAM8-22 binding modes simulated by Colabfold and in BAM8-22-MRGPRX1-Gq complex structure that our resolved. The BAM8-22 model 1-5 predicted by Colabfold are shown in plum, light cyan, light sky blue, tan and light coral, BAM8-22 in our resolved BAM8-22-MRGPRX1-Gq complex is shown in red. **(d)** Emax effects of different mutations within the ligand-binding pocket of MRGPRX1 on CNF-Tx2 induced G α i-G γ dissociation. Statistical differences between MRGPRX1 WT and mutations were presented as the mean \pm SEM of three independent experiments and determined by two-sided one-way ANOVA with Tukey test. *, P<0.05; **, P<0.01; ***, P<0.001; ns, no significant difference; ND, not detected. (P=0.2747, 0.0505, ND, ND, <0.001, 0.1296, ND, =0.999, 0.1859, <0.001 from left to right). Data from three independent experiments are presented as the mean \pm SEM (n=3). **(e-f)** Effects of different mutations within the ligand-binding pocket of MRGPRX1 on CNF-Tx2 induced G α i-G γ dissociation in MRGPRX1 overexpressing cells. The curve data from three independent measurements are measured as mean \pm SEM (n=3).

Fig. R4 (also shown in Supplementary Fig.12) Coupling of MRPGRX1 with Gi and Gq.

(a) Detailed interactions between the TM bundles of MRGPRX1 and the $\alpha 5$ -helix end of G αi . **(b)** Detailed interactions between the ICL2 of MRGPRX1 and the G αi . **(c)** Emax effects of different mutations of G protein interface mutations of MRGPRX1 on G αi . Statistical differences between MRGPRX1 WT and mutations were presented as the mean \pm SEM of three independent experiments and determined by two-sided one-way ANOVA with Tukey test. *, P<0.05; **, P<0.01; ***, P<0.001; ns, no significant difference; ND, not detected. (P=0.631, <0.1832, ND, ND, ND, <0.001, <0.3014, ND from top to bottom). Data from three independent experiments are presented as the mean \pm SEM (n=3). **(d)** The effects of G protein interface mutations of MRGPRX1 on G αi . The curve data from three independent measurements are measured as mean \pm SEM (n=3). **(e)** Detailed interactions between the bulky end of $\alpha 5$ helix of G αq and the V124^{3.54}, L198^{5.57}, R213^{6.32}, L214^{6.33} and T217^{6.36} of MRGPRX1. **(f)** Detailed interactions between the E357^{G.H5.22}, Y358^{G.H5.23} of G αq and Y64^{2.42}, F61^{2.39} and Y130^{ICL2}, R131^{ICL2} of MRGPRX1. **(g)** Emax effects of different mutations of G protein interface mutations of MRGPRX1 on G αq . Statistical differences between MRGPRX1 WT and mutations were presented as the mean \pm SEM of three independent experiments and determined by two-sided one-way ANOVA with Tukey test. *, P<0.05; **, P<0.01; ***, P<0.001; ns, no significant difference; ND, not detected. (P=0.999, <0.2713, 0.8716, <0.001, <0.001, <0.0781, 0.3189, <0.5425, ND, ND, <0.001 from top to bottom). Data from three independent experiments are presented as the mean \pm SEM (n=3). **(h)** The effects of G protein interface mutations of MRGPRX1 on G αq . The curve data from three independent measurements are measured as mean \pm SEM (n=3).

2. The new analysis of ligand bias (comment 4) is interesting but it seems only to be included as a figure for review. I think this should be included as a supporting figure unless there is a compelling reason to exclude it.

Reply: We thank the reviewer for his/her helpful suggestions. As suggested by the reviewer, we have added the new analysis of ligand bias to supplementary Fig. 10f-10h in the revised manuscript as follows:

Fig. R5 (also shown in Supplementary Fig. 11) Sequence alignment of peptide common motif recognized by MRGPRX1 and MRGPRX2.

(a) Peptide ligand sequences of MRGPRX2. Sequence comparisons of several peptide-based allergens with similarities to the $\phi^p(X_{0-1}) R/K^p(X_2) \phi^{p13}(X_{2-3}) \phi^{p16}(X_3) R/K^p(X_4)$ motif. (b) Peptide ligand sequences of MRGPRX1. Sequence comparisons of several peptide-based allergens with similarities to the $\phi^{B17}(X_{1-2}) R^{B20} \phi^{B21}$ motif. (c) Effects of BAM8-22, CNF-Tx2, γ 1-MSH, hemoglobin β -chain, P60 (part of C5orf29) induced MRGPRX1 activation evaluated via Gai-Gy dissociation assay. Data from three independent experiments are presented as the mean \pm SEM (n=3). All data were analyzed by two-sided one-way ANOVA with Turkey test. (d) Effects of different BAM8-22 mutations on BAM8-22 induced Gai-Gy dissociation. Data from three independent experiments are presented as the mean \pm SEM (n=3). All data were analyzed by two-sided one-way ANOVA with Turkey test. (e) Effects of different CNF-Tx2 mutations on CNF-Tx2 induced Gai-Gy dissociation. Data from three independent experiments

are presented as the mean \pm SEM (n=3). All data were analyzed by two-sided one-way ANOVA with Turkey test. **(f)** Concentration-dependent response curves of MRGPRX1 in response to different peptide or chemical ligands by G α i-G γ dissociation assay. Values are mean \pm SEM from three independent experiments (n=3) performed in triplicates. **(g)** Concentration-dependent response curves of MRGPRX1 in response to different ligands by G α q-G γ dissociation assay. Values are mean \pm SEM from three independent experiments (n=3) performed in triplicates. **(h)** Comparison of the biased properties of CNF-Tx2 and Chloroquine. Both CNF-Tx2 and Chloroquine were assessed for Gi signaling (f) and Gq signaling (g). The bias factor (β value) of CNF-Tx2 and Chloroquine was calculated using BAM8-22 as the reference.

3. There has been extensive analysis of protein-protein interactions (comment 7) applying modeling from AlphaFold2 in AlphaFold multimer (PMID: 35900023). Web tools such as Colabfold (<https://colab.research.google.com/github/sokrypton/ColabFold/blob/main/AlphaFold2.ipynb>) make this straightforward. This approach has been applied to other GPCR/ligand pairs (<https://pubmed.ncbi.nlm.nih.gov/37092865/>). I think a comparison of experimental data with AlphaFold2-based modeling results would be of wide interest.

Reply: We thank the reviewer for his/her helpful suggestions. As suggested by the reviewer, we have predicted the MRGPRX1-BAM and MRGPRX1-CNF interactions using Colabfold, the results have been added in supplementary Fig. 10a-10c. The corresponding description has been included in the section of “Binding of CNF-Tx2 to MRGPRX1” in the revised manuscript: *“Paralleling to the experimental determined structure, we have used Colabfold (see reference Mirdita et al, 2022, Nat Methods) (see reference Teufel et al, 2023, J Chem Inf Model) to predict the binding modes of CNF-Tx2 and BAM8-22 to MRGPRX1. we found that the ligand binding poses predicted by Colabfold were quite different from the binding patterns of the ligands in our resolved structures (Supplementary Fig. 10a-10c). We therefore speculated that the experimental data is still needed for analyzing the interaction between peptide ligand and their corresponding receptors.”*

Fig. R6 (also shown in Supplementary Fig. 10). Binding of CNF-Tx2 and BAM8-22 to MRGPRX1.

(a) Comparison of CNF-Tx2 binding modes simulated by Colabfold and in CNF-Tx2-MRGPRX1-Gi1 complex structure that our resolved. The CNF-Tx2 model 1-5 predicted by Colabfold are shown in tangold, skybluemedium slate blue, plummedium turquoise, cornflower bluelightgreen and dark salmonwheat, CNF-Tx2 in our resolved CNF-Tx2-MRGPRX1-Gi1 complex is shown in red. (b) Comparison of BAM8-22 binding modes simulated by Colabfold and in BAM8-22-MRGPRX1-Gi1 complex structure that our resolved. The BAM8-22 model 1-5 predicted by Colabfold are shown in plum, dark green, light sky blue, medium purple and dark orange, BAM8-22 in our resolved BAM8-22-MRGPRX1-Gi1 complex is shown in red. (c) Comparison of BAM8-22 binding modes simulated by Colabfold and in BAM8-22-

MRGPRX1-Gq complex structure that our resolved. The BAM8-22 model 1-5 predicted by Colabfold are shown in plum, light cyan, light sky blue, tan and light coral, BAM8-22 in our resolved BAM8-22-MRGPRX1-Gq complex is shown in red. **(d)** Emax effects of different mutations within the ligand-binding pocket of MRGPRX1 on CNF-Tx2 induced G α i-G γ dissociation. Statistical differences between MRGPRX1 WT and mutations were presented as the mean \pm SEM of three independent experiments and determined by two-sided one-way ANOVA with Tukey test. *, P<0.05; **, P<0.01; ***, P<0.001; ns, no significant difference; ND, not detected. (P=0.2747, 0.0505, ND, ND, <0.001, 0.1296, ND, =0.999, 0.1859, <0.001 from left to right). Data from three independent experiments are presented as the mean \pm SEM (n=3). **(e-f)** Effects of different mutations within the ligand-binding pocket of MRGPRX1 on CNF-Tx2 induced G α i-G γ dissociation in MRGPRX1 overexpressing cells. The curve data from three independent measurements are measured as mean \pm SEM (n=3).

Reviewer #2 (Remarks to the Author):

The authors have made an effort to address my concerns. However I still have two minor questions that should be addressed before supporting publication of the revised manuscript.

1. The local resolution map showed in Supplementary Fig 2d still seems incorrect and is not consistent with the Supplementary table 1 (map resolution range). The authors should check it carefully.

Reply: We thank the reviewer for his/her helpful suggestions. We have checked the local resolution map and the map resolution range carefully. The revised resolution and map resolution range in the Supplementary Fig 2d and those in the content in supplementary table 1 are now consistent. Notably, this is a collaboration work. The BAM8-22-MRGPRX1-Gq data are mainly handled by Dr. Zhang of Shanghai Institute of Materia Medica and we have several miscommunications in previous version, which lead to the observed inconsistency. The revised version of the map and table are listed as follows:

Fig. R7 (also shown in Supplementary Fig. 2) Cryo-EM images and single particle reconstruction of the BAM8-22-MRGPRX1-Gq complex.

(a) Flow chart for cryo-EM data processing of BAM8-22-MRGPRX1-Gq complex. **(b)** Representative Cryo-EM micrograph of BAM8-22-MRGPRX1-Gq complex (left) and 2D class averages (right). **(c)** Fourier shell correlation curves for the final 3D density maps of BAM8-22-bound MRGPRX1-Gq complex. At the Fourier shell correlation (FSC) 0.143 cut-off, the overall resolution was 2.7 Å. **(d)** 3D density map colored according to local resolution (Å) of the BAM8-22-MRGPRX1-Gq trimer complex.

Supplementary Table 1: Cryo-EM Data Collection, Model Refinement, and Validation

Statistics.

	BAM8-22- MRGPRX1-Gq (EMDB-36232) (PDB 8JGF)	CNF-Tx2- MRGPRX1-Gi (EMDB-36229) (PDB 8JGB)	BAM8-22- MRGPRX1-Gi (EMDB-36233) (PDB 8JGG)
Data collection and processing			
Magnification	81,000	130,000	130,000
Voltage (kV)	300	300	300
Electron exposure (e-/Å ²)	50	50	60
Defocus range (µm)	-1.2 to -2.2	-0.8 to -1.2	-0.8 to -1.2
Pixel size (Å)	1.04	1.08	0.89
Symmetry imposed	C1	C1	C1
Initial particle images (no.)	11,127,531	146,824,9	3,628,139
Final particle images (no.)	1,316,443	315,448	925,644
Map resolution (Å)	2.7	2.8	3.0
FSC threshold	0.143	0.143	0.143
Map resolution range (Å)	2.0-3.5	2.0-3.5	2.0-3.5
Refinement			
Initial model used (PDB code)	7UVY	7UVY	7UVY
Model resolution (Å)	3.1	3.0	3.2
FSC threshold	0.5	0.5	0.5
Model resolution range (Å)	2.0-3.5	2.0-3.5	2.0-3.5
Map sharpening B factor (Å ²)	-129.7	-101.8	-137.2
Model composition			
Non-hydrogen atoms	8338	7807	7889
Protein residues	1101	1062	1076
Ligands	1	1	1
B factors (Å ²)			
Protein	27.93	50.79	92.83
R.m.s. deviations			
Bond lengths (Å)	0.008	0.010	0.007
Bond angles (°)	1.105	1.384	1.145
Validation			
MolProbity score	1.78	2.16	1.92
Clashscore	8	7	9
Poor rotamers (%)	0.1	0	0.7
Ramachandran plot			
Favored (%)	95.74	93.82	94.30
Allowed (%)	4.26	6.18	5.70
Disallowed (%)	0	0	0

2. The resolution of BAM8-22- MRGPRX1-Gq provided in validation report is 2.7Å which is different from what was mentioned in the manuscript. And the validation reports of these three structures here are still informal version.

Reply: We thank the reviewer for his/her helpful suggestions. As mentioned above, we have carefully checked the calculation process of cryo-electron microscope data of BAM8-22-MRGPRX1-Gq complex and determined that the final resolution of the map should be 2.7 Å. Now, the description of the manuscript, supplementary figure 2, supplementary table 1 and the formal version of validation report are all consistent. The formal version of the three validation reports have been included in the revised version (PDB numbers of BAM8-22-MRGPRX1-Gq, CNF-Tx2-MRGPRX1-Gi and BAM8-22-MRGPRX1-Gi complex are 8JGF, 8JGB and 8JGG, respectively. EMD numbers of BAM8-22-MRGPRX1-Gq, CNF-Tx2-MRGPRX1-Gi and BAM8-22-MRGPRX1-Gi complex are EMD-36232, EMD-36229 and EMD-36233, respectively.). The validation reports of BAM8-22-MRGPRX1-Gq, CNF-Tx2-MRGPRX1-Gi and BAM8-22-MRGPRX1-Gi complex were generated by wwPDB OneDep System of RCSB Protein Data Bank on July 13, June 29 and July 2, respectively.